# Structural obstruction to full DNA replication in terminally differentiated skeletal muscle cells

Sara Zribi[1], Gladio Joel Giannitelli [ID][1], Laura Bernardini[2], Federica Censi [ID][3], Serena Camerini[1], Lucia Bertuccini [ID][1], Francesca Iosi [ID][1], Emilia Giannella[1], Massimo Sanchez[1], Elisa Consoli [ID][1], Andrea Casagrande [ID][4], Alessandra Cenci [ID][1], Giovanna Floridia [ID][3,8], Antonio Novelli [ID][2], Alessandro Giuliani [ID][5], Vincenzo Costanzo [ID][6,7], Marco Crescenzi [ID][1][✉] & Deborah Pajalunga[4]

## Abstract

**Terminal cell differentiation is often associated with permanent withdrawal from proliferation, termed the postmitotic state. Though widespread among vertebrates and determinant for their biology, the molecular underpinnings of this state are poorly understood. Postmitotic skeletal muscle myotubes can be induced to reenter the cell cycle; however, they generally die as a result of their inability to complete DNA replication. Here, we explore the causes of such incompetence. Genomic hybridization of newly synthesized DNA shows that the replicative failure does not concern specific genomic regions, but can stochastically affect any of them. Myoblast and myotube nuclei are incubated in replicative *Xenopus* egg extract, which provides a full DNA replication machinery. While myoblast nuclei attain complete DNA replication, those from myotubes, even in these conditions, duplicate less than half of their genomes, strongly indicating that the structure of myotube chromatin obstructs DNA replication. Furthermore, disassembling and disorganizing chromatin with a strong salt treatment does not modify the replicative differences between the two types of nuclei, suggesting that they are rooted in the core structure of chromatin.**

**Keywords** Cell Cycle; Chromatin; Histones; Terminal Differentiation; Xenopus
**Subject Categories** Chromatin, Transcription & Genomics; Musculoskeletal System; Stem Cells & Regenerative Medicine

## Introduction

A cell that has reached the final stage of specialization in its lineage is labeled terminally differentiated. In many cell types, terminal differentiation is associated with, and in fact requires, permanent exit from the cell cycle, a condition named postmitotic (PM) state (Ruijtenberg and van den Heuvel, 2016). Despite decades of efforts, the fundamental mechanisms underlying the PM state have not been fully elucidated: we still do not know what makes PM cells unable to proliferate.

One way to probe how the PM state is established and maintained is to reactivate the cell cycle in a suitable model system and evaluate the ensuing cellular and molecular responses. Investigating the PM state and its determinants has a twofold purpose. One is to address a basic question that widely concerns vertebrate biology. In addition, rescuing proliferation in PM cells in a controlled, safe, and reversible fashion might open new avenues for regenerative medicine (Blau and Pomerantz, 2011).

Skeletal muscle cells represent a prototypic PM cell type, that has often served as a model to investigate the PM state (reviewed in (Pajalunga and Crescenzi, 2021)). Such cells proliferate as myoblasts (MBs) and undergo terminal differentiation in vitro, by fusing into syncytial myotubes (MTs).

Reactivation of the cell cycle in MTs is attainable by a variety of means (reviewed in (Pajalunga and Crescenzi, 2021)). A short, non-exhaustive list includes infection with DNA tumor viruses (Fogel and Defendi, 1967; Yaffe and Gershon, 1967; Crescenzi et al, 1995), overexpression or suppression of cell cycle regulators (Latella et al, 2001; Pajalunga et al, 2007), repression of differentiation (Mastroyiannopoulos et al, 2012), and manipulation of the apoptotic process (Wang et al, 2015). Disappointingly, in most if not all instances, MTs do not display enduring cell proliferation, as the reactivated cells undergo apoptosis (Endo and Nadal-Ginard, 1998; Latella et al, 2000), cell cycle arrest (Novitch et al, 1996; Latella et al, 2001), or mitotic catastrophe (Novitch et al, 1996). On the contrary, when the same methods are used to reactivate the cell cycle of reversibly quiescent cells (Biferi et al, 2015) or even senescent ones (Falcone et al, 2013), they undergo sustained proliferation and suffer no damage (Pajalunga et al, 2010). This difference has found at least one explanation: upon cell cycle reactivation, regardless of the methods employed, MTs do undergo DNA replication to variable extents, but never complete it,

[1]Core Facilities, Italian National Institute of Health, Rome, Italy. [2]Medical Genetics Division, Casa Sollievo della Sofferenza Foundation, San Giovanni Rotondo, FG, Rome, Italy. [3]National Centre for Rare Diseases, Italian National Institute of Health, Rome, Italy. [4]Department of Oncology and Molecular Medicine, Italian National Institute of Health, Rome, Italy. [5]Environment and Health Department, Italian National Institute of Health, Rome, Italy. [6]IFOM ETS, The AIRC Institute of Molecular Oncology, Milan, Italy. [7]Department of Oncology and Hematology-Oncology, University of Milan, Milan, Italy. [8]Present address: Bioethics Unit, Italian National Institute of Health, Rome, Italy. [✉]E-mail: marcosilvia86@alice.it

leading them to inevitable proliferation failure (Pajalunga et al, 2010). The stark contrast between MTs and non-PM cells indicates that, in the former, DNA synthesis is intrinsically hindered by unknown mechanisms.

In principle, the potential causes of incomplete DNA synthesis can be subdivided into two non-mutually exclusive classes. The first one would include functional alterations: any modified or repressed biomolecular or biochemical mechanisms impinging on DNA replication. The second class might consist of physical obstacles that hinder full DNA replication, such as an impervious chromatin structure.

Here, we look for evidence of structural obstacles by exploiting in vitro replicative Xenopus egg extracts (XEE), which provide all the necessary and sufficient factors to replicate the DNA of exogenously added nuclei (Blow and Laskey, 2016). In this system, any functional alterations would be irrelevant, since the extract completely replaces the native DNA replication machinery of the added nuclei. Alternatively, the presence of structural hurdles would impair DNA synthesis even in this setting. Strikingly, we find that, in XEE, nuclei from MTs replicate less than half of their DNA, while MB nuclei fully duplicate theirs. Our results strongly indicate that MTs carry yet undiscovered physical obstacles to full DNA replication.

# Results and discussion

## A partial but uniformly distributed obstacle to DNA replication in MTs

Despite prolonged investigations, we have found little evidence of functional obstacles to DNA synthesis in MTs. A notable exception is an unbalanced and partially defective expansion of the nucleotide pool upon cell cycle reactivation (Pajalunga et al, 2017). Although restoring nucleotide balance in reactivated MTs (rMTs) improved DNA synthesis, it fell short of enabling its completion, suggesting that there might be other hurdles to full DNA replication. Thus, we chose to identify DNA regions that might be physically non-permissive to DNA replication.

For this purpose, we reactivated the cell cycle in MTs by RNA interference for the cyclin-dependent kinase inhibitors Cdkn1a (p21) and Cdkn1b (p27) (Pajalunga et al, 2007) in the presence of 20 μM 5-bromo-2′-deoxyuridine (BrdU) (Fig. EV1A,C). After DNA extraction and purification from the cells, the newly synthesized DNA was isolated by CsCl gradient centrifugation. It was then analyzed by comparative genomic hybridization (CGH, Fig. 1A), relative to a 2n reference, constituted by an equal amount of DNA from exponentially proliferating MBs uniformly labeled on one strand (i.e., hemisubstituted) with BrdU. If in the DNA of rMTs there were specific hypo- or non-replicating regions, they would be underrepresented. In contrast, Fig. 1B,C shows that all genomic regions were very similarly represented in the neoreplicated DNA from the two cell types: in two independent experiments, the log$_2$ ratios of the amounts of DNA for all regions in the two cell types oscillated tightly around 0. We stress that, in CGH experiments, we loaded equal amounts of BrdU-hemisubstituted DNA from both rMTs and reference MBs. Since the former replicate markedly less DNA, the required amount was obtained from a larger number of rMT than MB genomes (Fig. 1A). Thus, the flat line in Fig. 1B,C indicates that all regions are similarly

represented in the two samples, not that the two cell types replicate their DNA to the same extent.

As a control, MBs rendered quiescent in suspension culture were synchronously reactivated by replating and labeled with BrdU (Fig. EV1B,C). The cells were harvested in mid-S phase and compared with freely proliferating MBs as described above. The reactivated MB DNA exhibited the expected enrichment in early-replicating regions and a corresponding depletion in late-replicating regions (Fig. 1D,E).

Hence, in rMT no region was totally or specifically unamenable to replication. Rather, all regions had the same, less-than-full probability of being replicated or not. This is consistent with immunofluorescence staining of rMT for BrdU, where different nuclei present marked disparities in the intensity and distribution of fluorescence, even within the same myotube (e.g., Pajalunga et al, 2010).

These results suggest a hindrance to DNA replication that, though not absolute, is uniformly distributed throughout MT chromatin.

## A structural obstacle to DNA replication in MTs

To assess whether DNA replication in rMTs is constrained by functional alterations or physical obstruction (see Introduction), we exploited XEE. Permeabilized nuclei can be incubated in these replicative extracts to induce DNA synthesis. Since XEE provides all the biochemical machinery for DNA replication, this is carried out independently of endogenous nuclear factors. In fact, most of the latter are lost, due to nucleus permeabilization.

Nuclei were isolated from proliferating MBs or MTs and permeabilized with Triton X-100. To ascertain that nuclei from both cell types were similarly accessible to macromolecules, they were incubated with FITC-labeled dextran sulphate (average molecular weight: 150 kDa). Figure EV2 shows that dextran entered MB and MT nuclei to an equal extent.

Next, permeabilized nuclei from quiescent MBs, proliferating MBs, and MTs were incubated in XEE, in the presence of $^3$H-dTTP. At regular intervals, aliquots were harvested from the reactions and the radioactivity incorporated into DNA was measured in a scintillation counter (Fig. 2A). In three independent experiments summarized in Fig. 2B, MB nuclei promptly began to synthesize DNA, while MT nuclei displayed a lag of about 1 h before DNA replication started. Most important, there was a marked difference in the amount of incorporated $^3$H-dTTP between MTs and MBs. As expected, nuclei from proliferating MBs synthesized less DNA than those from quiescent ones, since some of the former had already undergone S phase, completely or in part, and could not reduplicate already-replicated regions (Arias and Walter, 2005). However, even nuclei from proliferating MBs incorporated distinctly more radioactive label than MT nuclei, in three independent experiments (presented separately in Fig. EV3). Yet, we underscore that nuclei from MTs are best compared with those from quiescent MBs since, initially, both contain only DNA that has not undergone replication in vivo. The data obtained with MT nuclei parallel those obtained with cultured rMTs (Pajalunga et al, 2017). This suggests that the XEE system encounters the same obstacle(s) to full DNA synthesis that is observed in vivo. As a control, we show that purified, high-molecular-weight DNA is replicated identically in XEE, whether it is extracted from MBs or MTs (Fig. EV4).

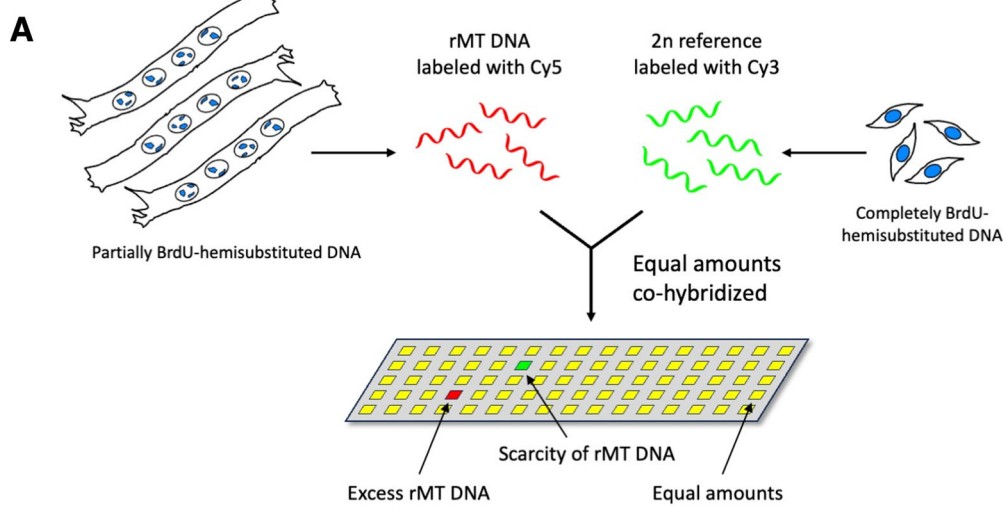

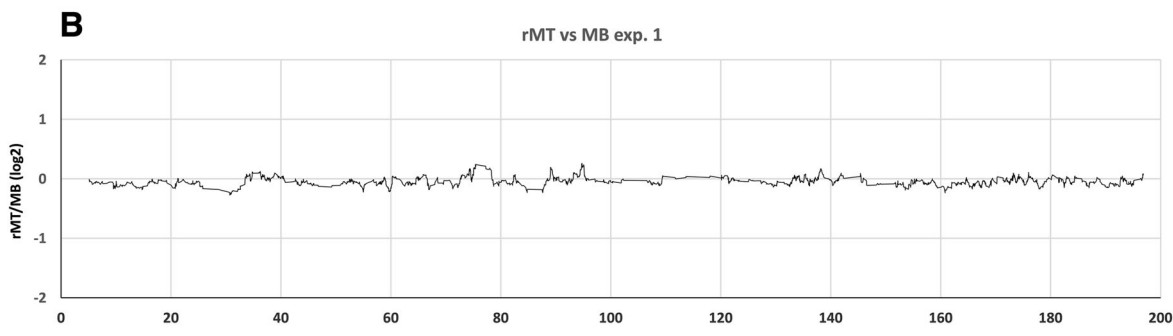

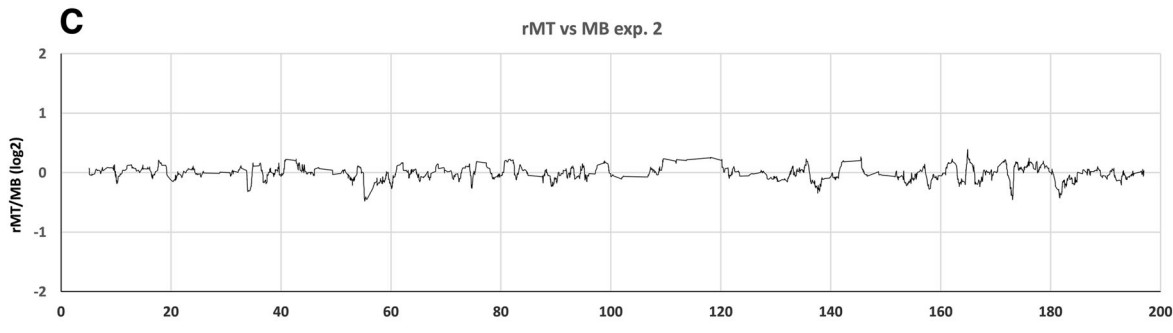

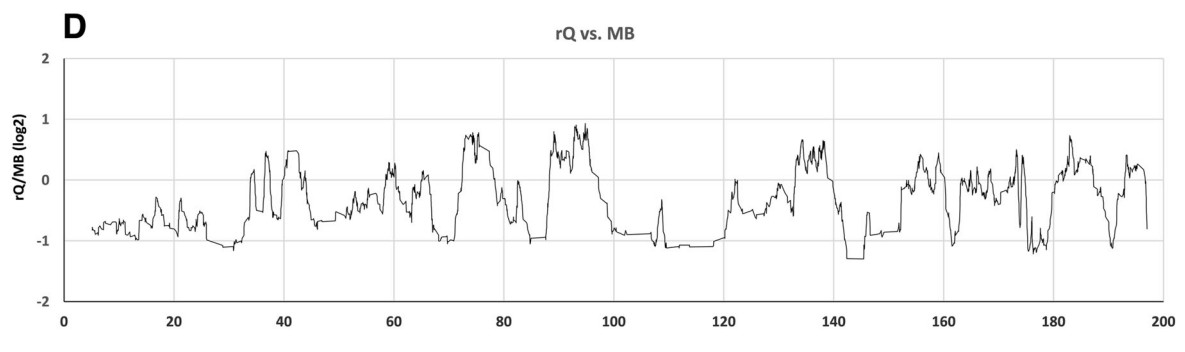

Position on chromosome 1 (Mbp)

**Figure 1. Comparative genomic hybridization.**

(A) Schematic of comparative genomic hybridization. The test (rMT) and reference (MB) DNA are labeled with Cy5 and Cy3, respectively. Equal amounts of both are mixed and hybridized onto a chip containing oligonucleotide probes for the entire mouse genome, except for highly repetitive regions. If the test DNA is overrepresented in a given region, compared to the reference DNA, the corresponding spots will exhibit higher red than green fluorescence, and vice versa. (B, C) Newly synthesized, BrdU-hemisubstituted DNA was isolated from MTs reactivated by RNA interference for p21 and p27 (rMT) or exponentially growing MBs, labeled with Cy3 or Cy5, respectively, and hybridized onto genomic mouse chips. $Log_2$ of signal ratios (Cy5/Cy3) are plotted against position on mouse chromosome 1. A value of 0 means that there is no difference in the intensities of the two signals (Cy3/Cy5 = 1; $log_2$ 1 = 0). The curve has been smoothed using a moving average, with a period of 20. The A and B panels represent two independent replicas of the same experiment. MT reactivation efficiency was 30% in the first experiment and 28% in the second. These percentages do not affect the results, since we isolated BrdU-hemisubstituted DNA, thus considering only reactivated MTs. (D) MBs, made quiescent by anchorage deprivation (culture on agar plates) were synchronously reactivated (rMT) by plating them into tissue culture dishes and harvested in mid-S phase. Their BrdU-hemisubstituted DNA was compared with that of exponentially growing MBs as above. (E) For reference, the DNA replication timing of chromosome 1 in fetal MBs (a cell type similar to, but distinct from our adult MBs) is shown. Data from Table S2 of (Hiratani et al, 2010) have been visualized using the IGV viewer (Robinson et al, 2023). Source data are available online for this figure.

Altogether, we interpret these results as a strong indication that MTs cannot completely replicate their DNA owing, at least in part, to the physical configuration of their nuclei.

## Preparation and characterization of partially deproteinated nuclei

We hypothesized that chromatin proteins, or less likely other nuclear macromolecules, might hinder DNA replication. To begin to characterize relevant factors, we sought to narrow the range of candidate molecules. We reasoned that we should reduce the protein content of both MT and MB nuclei, provided that they retained their difference in DNA replicability.

To this aim, we elected to treat Triton-permeabilized MT nuclei with high-molarity NaCl, to break weaker intermolecular bonds and detach proteins not strongly bound to DNA or chromatin. In preliminary experiments, we treated the nuclei with NaCl concentrations ranging from 0.1 to 0.7 M. The suspensions were then centrifuged to separate solubilized proteins, which diffuse from the permeabilized nuclei into the supernatant, from those retained within the nuclei. We then analyzed three proteins by western blotting in both the supernatants and pellets: pRb as an example of a non-histone protein strongly attached to chromatin (Mittnacht and Weinberg, 1991), H1 as the histone most loosely bound to DNA, and H3 as a representative of core histones, firmly tied to DNA. Figure EV5 shows that pRb and histone H1 were progressively solubilized with increasing salt concentrations, while histone H3 largely remained in the nuclei. On the basis of these results, we chose 0.5 M NaCl for subsequent experiments.

To assess to what extent salt treatment affected chromatin integrity and conformation, we incubated permeabilized nuclei from MBs and MTs in 0.5 M NaCl and analyzed them by transmission electron microscopy and mass spectrometry. Electron microscopy showed that in both cell types, chromatin was arranged in denser clumps and lighter areas, indicative of hetero- and euchromatin conformation, respectively (Fig. 3A,B,E,F). In sharp contrast, NaCl-treated nuclei from MBs and MTs displayed a far more homogeneous distribution of chromatin, with no major variations in electron density. This effect is most clearly evident in the nucleoli, which are very conspicuous in the controls, but nearly disappear in salt-treated nuclei (Fig. 3C,D,G,H). Thus, as expected, our salt treatment disrupts most intermolecular interactions and the nuclear structures that depend on them.

To further characterize the effects of the salt treatment, we analyzed nuclear protein extracts by mass spectrometry. Four independent samples of permeabilized MT nuclei, mock-treated or incubated with 0.5 M NaCl, were subjected to mass spectrometry. In each sample, all protein intensity values were normalized to the total intensity of core histones (Dataset EV1). Altogether, 1114 nuclear proteins were quantitated in both treated and control nuclei. Figure 4A,B shows that, in NaCl-treated nuclei, 79% of the nuclear proteins were decreased, on average by 40%, compared with controls. Hence, as already indicated by electron microscopy, salt treatment strongly reduces total nuclear protein content and, specifically, chromatin proteins that might interfere with DNA replication. In fact, even 83% of linker histone H1 was displaced from salt-treated chromatin (Dataset EV1, rows 1424–1428).

Interestingly, NaCl treatment nearly halved the CTCF protein content in salt-treated, compared with control nuclei. Similarly, salt reduced STAG1 content to 47%. The STAG2 protein was lowered to 70%; although this last result was not statistically significant, the reduction was consistent across four experiments (Fig. 4C; Dataset EV1). CTCF is a core architectural protein that contributes to establish DNA loops and the three-dimensional organization of the genome. The STAG proteins are subunits of the Cohesin complex, which is critically involved in similar functions and essential for CTCF activity at most genomic sites (Ong and Corces, 2014). These data will help to interpret the results described in the following paragraph.

## Partial deproteination of MT nuclei does not alter their resistance to DNA replication

We wished to determine whether salt-treated MT nuclei are more permissive to DNA replication than controls. To this aim, we incubated in XEE MB and MT nuclei, treated with NaCl or not, and assessed total DNA content at 0, 1, 2, and 4 h. In these experiments, we did not use radioactively labeled nucleotides. Instead, the nuclei were incubated with 4′,6-diamidino-2-phenylindole dilactate (DAPI) and their relative DNA content was measured by microfluorimetry. This method enables the measurement of DNA content in individual nuclei, rather than the whole population. In these experiments, nuclei from quiescent MBs were not included, because remnants of the agar used to culture them in suspension (see "Methods") interfered with accurate DNA quantitation by microfluorimetry. Figure 5A summarizes the results of two independent experiments and shows that MB nuclei nearly doubled their median DNA content within 4 h. In sharp contrast, the DNA content of MT nuclei increased by less than 50%, in good

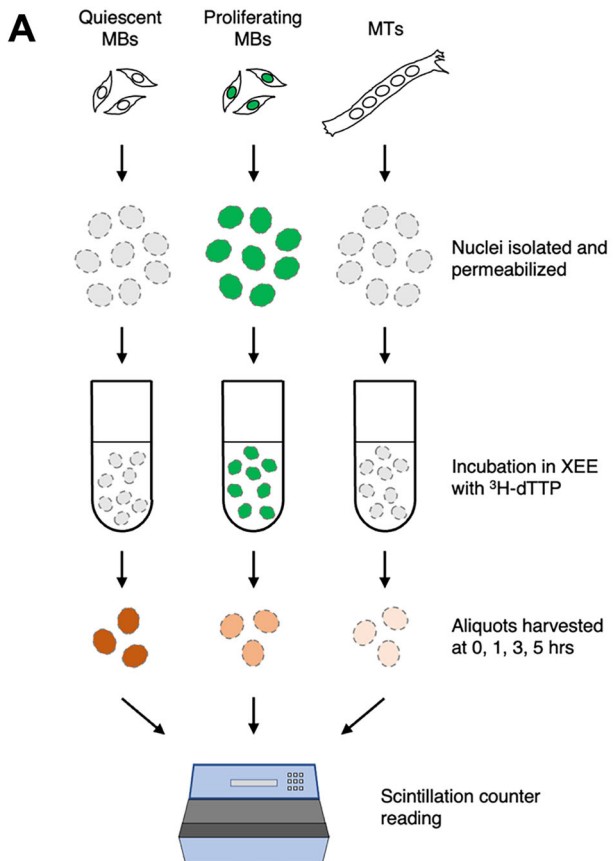

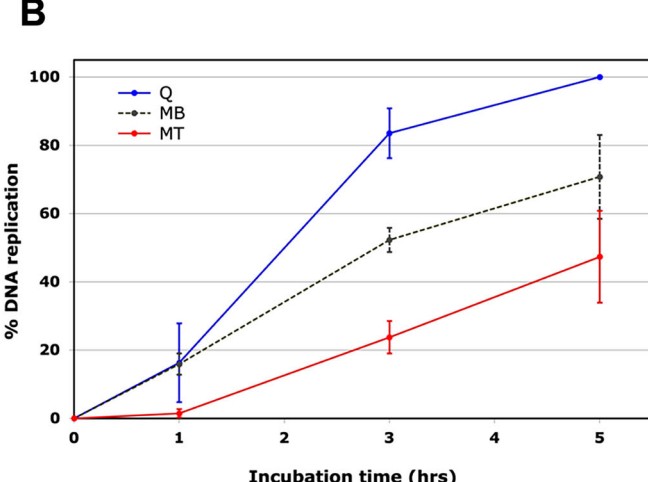

**Figure 2. DNA replication in nuclei incubated in XEE.**

(A) Schematic of the experiment. Isolated, permeabilized nuclei from quiescent or proliferating MBs and MTs were incubated in XEE in the presence of tritiated dTTP. At the indicated times, aliquots were harvested from the incubation mix and the radioactive label incorporated into DNA was measured with a scintillation counter. (B) Three independent experiments were averaged and are shown with standard deviations. In each experiment, the 5 h reading of Q was equaled to 100% and all other points expressed as percentages of that value. Source data are available online for this figure.

agreement with the results in Fig. 2 (note that Fig. 2 displays newly synthesized DNA, while Fig. 5 shows total DNA content). In addition, in MT nuclei, DNA synthesis began only after about 1 h, confirming the results presented in Fig. 2. Strikingly, salt treatment did not alter DNA synthesis in either MBs or MTs, showing that the loss of nuclear proteins attained was irrelevant to the difference in DNA replicability between the two types of nuclei. Pictures of the nuclei measured by microfluorimetry are shown in Fig. EV6. The quantitative results described above are paralleled by nuclear morphology: while MB nuclei enlarge within the first hour in XEE, MT ones remain unchanged during the same time. After 4 h, MB nuclei are deformed and diaphanous, whereas those of MTs are only slightly enlarged. Again, these morphological features are not affected by salt treatment.

A different view of the same data is presented in Fig. 5B, which displays the percentages of nuclei whose DNA content falls within each of 16 equally wide intervals on an arbitrary scale. At time 0, the main DNA content peak was 12.3 in all cases. However, MBs showed a "shoulder", peaking at 17.9, reflecting a subpopulation of nuclei in $S/G_2$, which was absent in MTs. MBs essentially doubled their DNA content by 4 h, with a 23.6 DNA content peak at that time (+91.9%, compared with time 0), while the corresponding MT peak was 17.9 (+45.5%). Again, the behavior of both MB and MT nuclei was not modified by NaCl treatment. This indicates that most nuclear proteins, whose levels were reduced by salt with no influence on DNA synthesis, do not play a role in restricting DNA replication in MTs.

Figure 5B also shows that some XEE-incubated MB nuclei underwent endoreduplication (DNA content higher than 4n), which was far less conspicuous in MT nuclei. Once more, this behavior was indifferent to salt treatment. To gain insight into the dynamics of DNA replication in the four populations considered, we pulsed them for 30' with fluorescent Cy3-dCTP and determined the percentages of nuclei performing DNA synthesis at various time points (Fig. EV7). Once more, MT nuclei performed very differently from those of MBs, whether quiescent or proliferating. The percentage of Cy3-positive MB nuclei was maximal between 1 and 3 h, dropped drastically between 3 and 7 h, then rose again at 9 h. This evidence is consistent with MB nuclei completing DNA synthesis between 3 and 7 h and ceasing replication. However, a significant proportion of MB nuclei underwent endoreduplication between 7 and 9 h, or even earlier. The percentage of Cy3-positive MT nuclei also peaked at 3 h, then began decreasing gradually, never to rise again within the time frame of the experiment. The simplest interpretation of these data, together with those of Fig. 5B, is that MT nuclei either ceased DNA replication before it was completed or continued to synthesize DNA slowly, rarely attaining a 4n equivalent.

Altogether, these experiments indicate that incomplete DNA replication in MT nuclei is most likely not due to the many nuclear proteins whose abundance has been significantly reduced by the NaCl treatment. Likewise, chromatin arrangement, dramatically disrupted by salt treatment (Fig. 3), cannot account for the impairment in DNA synthesis that characterizes MTs. Further evidence is provided by the fact that NaCl treatment caused a reduction in the levels of CTCF, STAG1, and possibly of STAG2, key proteins in establishing and maintaining the three-dimensional organization of the genome (see above and Fig. 4C), yet it did not modify DNA replication in XEE. Finally, DNA loop structures typically range from 100 to 250 kb in size (Sept et al, 2025);

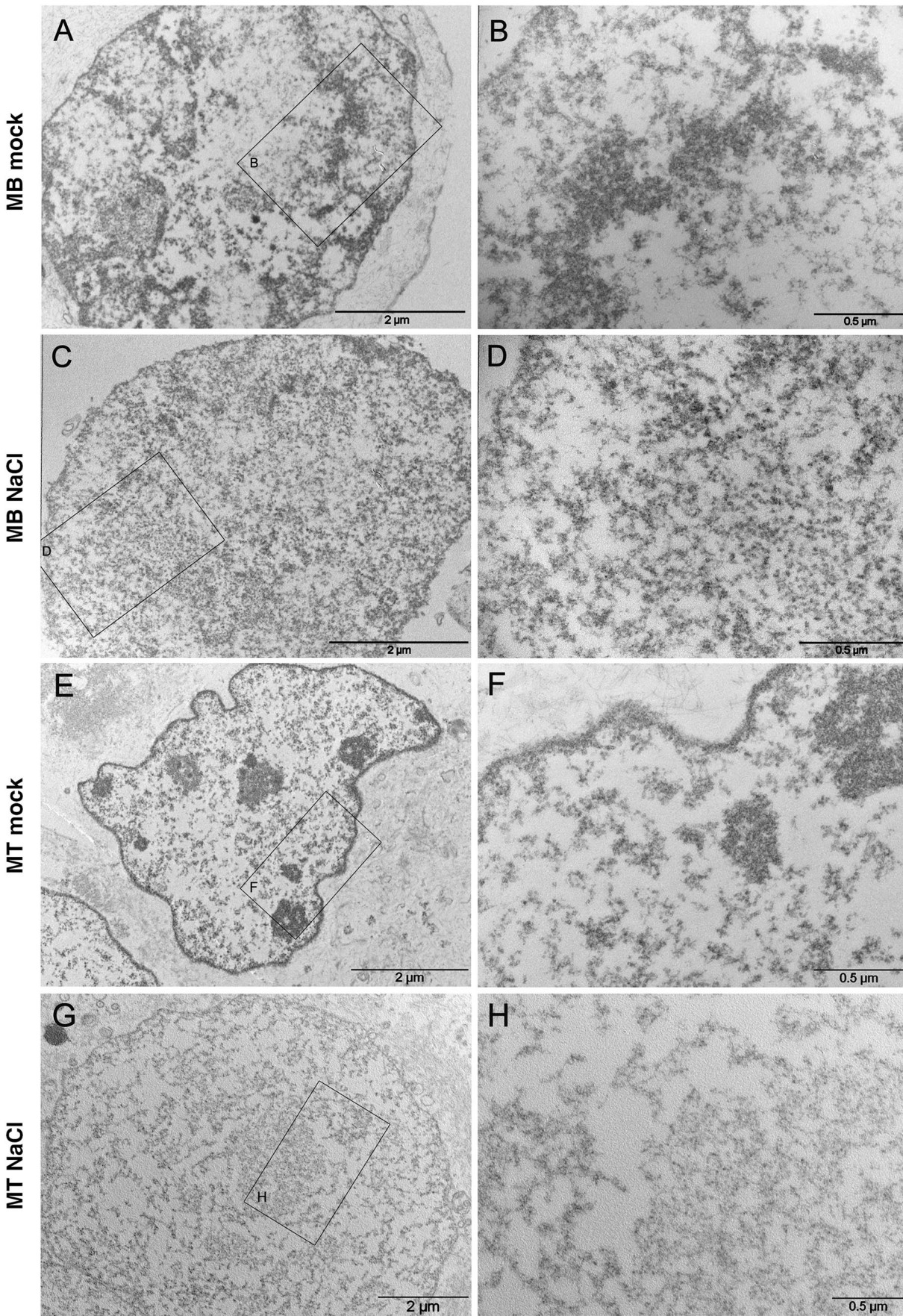

**Figure 3. Electron microscopy of nuclei.**

Nuclei isolated from MBs and MTs were treated with 0.5 M NaCl or mock-treated, as indicated. The nuclei were then imaged by transmission electron microscopy. Right-side panels (**B, D, F, H**) are magnifications of the areas outlined in the corresponding left-side panels (**A, C, E, G**). Source data are available online for this figure.

however, our CGH analysis, with a resolution of 50 kb, shows no evidence of their involvement in the incomplete DNA replication observed in myotubes (i.e., no detectable underreplicated regions).

In this paper, we confirm that PM MTs are intrinsically unable to completely duplicate their DNA, upon cell cycle reactivation (Pajalunga et al, 2010). Remarkably, our results strongly indicate that this inability should be ascribed primarily to structural obstacles, as opposed to functional failures. Indeed, defective DNA replication cannot be rescued by the complete replication machinery provided by XEE, even in the greatly simplified context of isolated nuclei.

Our data generate a partial but suggestive identikit of the molecules underlying such obstacles. (i) They should be abundant and evenly dispersed throughout MT chromatin, because every genomic region has the same (reduced) probability of being replicated as any other one (Fig. 1). Hence, they represent not an absolute obstacle, but one that produces apparently stochastic effects, at least at the resolution level of our CGH (50,000 bp). (ii) They should be strongly attached to chromatin, as salt-mediated, vigorous nuclear deproteination does not improve DNA synthesis even slightly. (iii) Their effects are probably not mediated by chromatin conformation or arrangement, which are both severely disrupted by our salt treatment.

In our view, this description best applies to nucleosome proteins. We would like to suggest that, in MTs, a peculiar composition of nucleosomes with respect to core histone variants and/or specific combinations of histone posttranslational modifications might render MT chromatin partially nonpermissive to replication. Thus, the current work presents a fresh perspective on the resistance to DNA synthesis of rMTs, largely as a consequence of structural features. At the same time, partial nucleus deproteination restricts the number of candidate determinants of such resistance, thus greatly simplifying the quest for their eventual identification. Further studies will be needed to pinpoint such determinants.

Finally, MTs can be regarded as a model system for terminal differentiation and indeed share responses to cell cycle reactivation with other PM cell types (e.g., Latella et al, 2001). We have argued in the past that other PM cells, when forcibly induced to reenter the cell cycle, appear to undergo only partial DNA duplication (Pajalunga et al, 2010). Thus, the conclusions of the present work potentially apply to a wide range of PM cells.

## Methods

### Reagents and tools table

| Reagent/resource | Reference or source | Identifier or catalog number |
| --- | --- | --- |
| **Experimental models** | | |
| RbLox 129/FVB mice | Gift of Anton Berns | Marino et al, 2000 |

| Reagent/resource | Reference or source | Identifier or catalog number |
| --- | --- | --- |
| **Antibodies** | | |
| IgG anti-pRb | BD Pharmingen | 554136 |
| IgG anti-H3 | Cell Signaling Technology | 9715 |
| IgG anti-H1.0 | Santa Cruz Biotechnology | sc-67324 |
| Goat anti-mouse IgG-HRP | Jackson ImmunoResearch | 115.035.003 |
| Goat anti-rabbit IgG-HRP | Bio-Rad | 310006678 |
| **Oligonucleotides and other sequence-based reagents** | | |
| siRNA Cdkn1a (p21) | Dharmacon | sequence: GGUGAUGUCCGACCUGUUC |
| siRNA Cdkn1b (p27) | Dharmacon | sequence: GAGAAGCACUGCCGGGAUA |
| **Chemicals, enzymes, and other reagents** | | |
| HCl | Sigma-Aldrich | 30721 |
| KCl | J.T. Baker | 0509 |
| NaCl | Sigma-Aldrich | 31434 |
| NaOH | Panreac | 131687.1211 |
| $MgSO_4$ | Panreac | 141404.1211 |
| $MgCl_2$ | Panreac | 131396.1210 |
| $CaCl_2$ | Panreac | 211221.1211 |
| $H_3PO_4$ | Normapur | 20 624.295 |
| $OsO_4$ | Electron Microscopy Science | 19110 |
| CsCl | Sigma-Aldrich | 289329 |
| Methanol | VWR, BDH Chemicals | 20864.320 |
| Ethanol | VWR, BDH Chemicals | 153385E |
| Phenol | Sigma-Aldrich | 108-95-2 |
| Potassium acetate | Riedel - de Haën | 32309 |
| Formic acid | J.T. Baker | 6037 |
| Tannic acid | Sigma-Aldrich | 403040 |
| Trichloroacetic acid (TCA) | Panreac | 131067.1609 |
| Acetonitrile | Sigma-Aldrich | 34888 |
| Trichloromethane | Panreac | 131252 |
| Phenylmethanesulfonyl fluoride (PMSF) | Sigma | P-7626 |
| DL ditiothreitol (DTT) | Sigma | D-9163 |
| 2-mercaptoethanol | J.T. Baker | 8683 |
| Formaldehyde | Sigma-Aldrich | 252549-IL |

| Reagent/resource | Reference or source | Identifier or catalog number |
|---|---|---|
| Paraformaldehyde | Electron Microscopy Science | 15710 |
| Glutaraldehyde | Electron Microscopy Science | 16210 |
| Ammonium bicarbonate | Riedel - de Haën | 11213 |
| Cycloheximide | Calbiochem | 66-81-9 |
| Iodoacetamide (IAM) | Sigma-Aldrich | I6125 |
| Sucrose | Sigma-Aldrich | 1604 |
| Glycerol | Panreac | 141339.1211 |
| Glycine | Sigma-Aldrich | 33226 |
| PBS | Biowest | L0615 |
| EDTA | Panreac | 131669.1210 |
| EGTA | MP Biomedicals | 195174 |
| HEPES | BDH Biochemical | 44147 |
| PIPES | Sigma-Aldrich | 5625-37-6 |
| Tris base | USB Corporation | 75825 |
| Sodium cacodylate buffer | TAAB Laboratories Equipment | S006 |
| Tris(2-carboxyethylphosphine) (TCEP) | Sigma-Aldrich | 646547 |
| Tween 20 | USB Corporation | 20605 |
| Triton X-100 | Eurobio | GAUTTR00-01 |
| SDS | Sigma-Aldrich | L4509 |
| Bromophenol blue | Sigma | B-6896 |
| Epoxy resin | Agar Scientific | AGR1031 |
| ATP | Sigma-Aldrich | 51963-61-2 |
| Inorganic pyrophosphate (PPi) | Sigma-Aldrich | 7722-88-5 |
| 5-bromo-2-deoxyuridine (BrdU) | Sigma-Aldrich | B5002 |
| 3H-methyl-thymidine, 20 Ci/mmol | PerkinElmer | NET027X001MC |
| 3H-methyl-thymidine, 20 Ci/mmol | PerkinElmer | NET027X001MC |
| Cy3-dCTP | GE Healthcare | PA53021 |
| Cy3 | Agilent Technologies | part of 5190-3399 |
| Cy5 | Agilent Technologies | part of 5190-3399 |
| DAPI, dilactate | Sigma | D9564 |
| Fluorescein-isothiocynate dextran (average MW 150 kDa) | Sigma-Aldrich | FD150S |
| Trypsin | Gibco | 15090-046 |
| Sequencing Grade Modified Trypsin | Promega | V5111 |
| Proteinase K | Sigma-Aldrich | 1.24568 |

| Reagent/resource | Reference or source | Identifier or catalog number |
|---|---|---|
| RNase A | Sigma | R-4875 |
| Nonfat milk | ITW Reagents | A30830 |
| Gelatin from porcine skin | Sigma | G2500 |
| Human chorionic gonadotropin | Sigma-Aldrich | 9002-61-3 |
| Calcium ionophore | Sigma | 52665-69-7 |
| Leupeptin | Sigma | 103476-89-7 |
| Cytochalasin B | Sigma | 14930-96-2 |
| Creatine phosphokinase | Sigma-Aldrich | C-3755 |
| Phosphocreatine, disodium salt hydrate | Sigma-Aldrich | P7936 |
| Penicillin/streptomycin | Biowest | L0022 |
| Chicken embryonic extract (CEE) | Prepared in-house, as described | N/A |
| Fetal bovine serum | Biowest | S1810 |
| Recombinant Human FGF-basic | Peprotech | 100-18B |
| *Xenopus laevis* eggs | Provided by Dr. Vincenzo Costanzo | N/A |
| Dulbecco's modified Eagle Medium (DMEM) | Life Technologies | L0103 |
| Ham's F10 Nutrient Mix, Glutamax supplemented | Life Technologies | 41550 |
| HiPerfect(TM) Transfection Reagent | Qiagen | 301707 |
| HALT(TM) Protease Inhibitor Cocktail | Thermo Scientific | 78430 |
| NuPage(TM) LDS Sample Buffer | Invitrogen | NP0007 |
| NuPage(TM) Sample Reducing Agent | Invitrogen | NP0009 |
| NuPage 4–12% Bis-Tris gel | Invitrogen | NP0321 |
| Nitrocellulose membrane | Cytiva | 10600016 |
| Lightwave PLUS ECL solution | GVS | LW0004 |
| GenomePlex(TM) Complete Whole Genome Amplification (WGA) Kit | Sigma-Aldrich | 41121800 |
| Sure Tag Genomic DNA labelling kit | Agilent Technologies | 5190-3399 |
| Wash buffers 1 and 2 | Agilent Technologies | 5188-5221 and 5188-5222 |
| Mouse Genome CGH Microarray 4x44K | Agilent Technologies | G4426B-015028 |
| Oligo aCGH/ChIP-on-Chip Hybridization Kit | Agilent Technologies | 5188-5220 |
| Whatman GF/C 2.5 glass microfilters | Camlab | 1822-025 |
| UA-Zero EM Stain | Agar Scientific | AGR1000 |

| Reagent/resource | Reference or source | Identifier or catalog number |
|---|---|---|
| Acclaim PepMap 100 C18 (trap column) | Thermo Fischer Scientific | 160454 |
| Silica Self Pack NanoLC column with emitter tip, 360 μm OD x 75 μM ID x 8 μm Tip x 50 cmL | MSWIL | ICT36007508-50-5 |
| ReproSil-Pur 120 C18-AQ, 1.9 μm | Dr. Maisch | r119.aq.0001 |
| Slide-A-Lyzer MINI Dialysis Devices, 10 K MWCO, 2 ml | Thermo Scientific | 88404 |
| **Software** | | |
| Feature Extraction v.8.0 | Agilent Technologies | |
| CytoGenomics v.2.5.8.11 | Agilent Technologies | |
| Adobe Photoshop CS5 | Adobe Systems | |
| Zen 3.3 Blue Edition | Zeiss | |
| ImageJ v.1.45 | https://imagej.net | |
| DIA-NN v.1.8.1 | https://github.com/vdemichev/DiaNN | |
| Perseus v. 2.0.11 | https://maxquant.net/perseus/ | |
| **Other** | | |
| Bandelin Sonopuls mini20 homogenizer | Bandelin | |
| ImageQuant LAS 4000 scanner | Cytiva Life Sciences | |
| G2545A hybridization oven | Agilent Technologies | |
| G5761A microarray scanner | Agilent Technologies | |
| LSM980 confocal microscope | Zeiss | |
| Axioskop 2 Plus fluorescence microscope | Zeiss | |
| AxioCam camera | Zeiss | |
| Neofluar 40x objective | Zeiss | |
| TLA100 rotor | Beckman Coulter | |
| VTi65 rotor | Beckman Coulter | |
| LS 6000 SC liquid scintillator | Beckman Coulter | |
| EM UC6 ultramicrotome | Leica Microsystems | |
| EM208S transmission electron microscope | Philips | |
| SIS Megaview II camera | Olympus | |
| Ultimate 3000 UHPLC | Thermo Fisher Scientific | |
| Orbitrap Fusion Tribrid mass spectrometer | Thermo Fisher Scientific | |

## Cells and cell cultures

MBs were originally isolated from RbLox 129/FVB mice (Marino et al, 2000). The cells did not undergo Cre-mediated Rb gene deletion and were used for continuity with previous experiments

(Pajalunga et al, 2010). Cells were seeded on gelatin-coated dishes (Sarstedt), prepared by covering the dish surface with a 0.06% gelatin solution, incubating at 4 °C for 30 min and then removing the solution and air-drying the dish. MBs were cultured in growth medium (Ham's F10 Nutrient Mix (Life Technologies), supplemented with 20% fetal bovine serum (Biowest), 3% home-prepared chicken embryonic extract (CEE), 2.5 ng/ml basic-FGF (Peprotech) and 1% penicillin/streptomycin (Biowest)), at 38 °C and 7% $CO_2$. Cells were detached from culture plates by incubation with PBS, supplemented with 0.5 mM EDTA for 10 min at 37 °C. Cells are routinely screened for mycoplasmas.

For CEE preparation, fertilized chicken eggs are incubated in a humidified incubator for 11 days, at 38 °C. After spraying the eggs with ethanol, the eggshells are cracked and the embryos are collected. Embryos are mashed by passing them through a needleless 50-mL syringe. The resulting mass is diluted with an equal volume of HAM-F10 medium and incubated for 30 min at RT with gentle shaking every 10 min. It is then centrifuged at 1400 g for 10 min. The supernatant (CEE) is collected, aliquoted, and stored at −80 °C. CEE is used without further purification.

Muscle differentiation was induced by seeding proliferating MBs onto gelatin-coated dishes in differentiation medium (high glucose DMEM (Life Technologies) supplemented with 10% fetal bovine serum and 1% penicillin/streptomycin). After 48 h, the vast majority of the cells were differentiated (differentiation index >95%).

MBs were made quiescent by culturing them in suspension on semi-solid medium (Milasincic et al, 1996) (growth medium with 0.7% Noble agar (Life Technologies)) for 21 h. Cells were induced to reinitiate proliferation by replating them onto gelatin-coated dishes in growth medium.

Cell cycle reactivation in myotubes was obtained by RNA interference for the cell cycle inhibitors Cdkn1a (p21) and Cdkn1b (p27). MTs cultured in 2 ml of differentiation medium in 35 mm gel-coated plates were transfected by adding to the culture medium 200 μl of DMEM, containing siRNA(s) (60 pM each) and 3% HiPerfect Transfection Reagent (Qiagen). Interference was allowed to proceed until analysis. Transfection efficiency is 100% if determined with fluorescent siRNAs. However, MT reactivation shows interexperimental variation (range: 25–80%).

When required, cells were fixed for 10 min at room temperature in 4% formaldehyde/PBS.

## Isolation of nuclei

Cells were harvested, centrifuged at $270 \times g$ for 10 min at 4 °C, then washed once in PBS. Pellets were either stored at −80 °C or used fresh for nuclear extraction.

To isolate nuclei, pellets were resuspended in Hypotonic Buffer (10 mM HEPES pH 7.5, 2 mM KCl, 2 mM $MgCl_2$, 3 mM EDTA, HALT Protease Inhibitors (Thermo Scientific), 1 mM 1,4-dithio-threitol (DTT)) and incubated on ice for 30 or 60 min (MBs or MTs, respectively). MBs were then centrifuged again and resuspended in PBS containing 25 μg/ml trypsin, then incubated at 25 °C for 2 min to release nuclei. MTs received a similar treatment, but were resuspended in 37.5 μg/ml trypsin and incubated for 4 min. To stop digestion, phenylmethanesulfonyl fluoride (PMSF) was added to a final concentration of 1 mM and

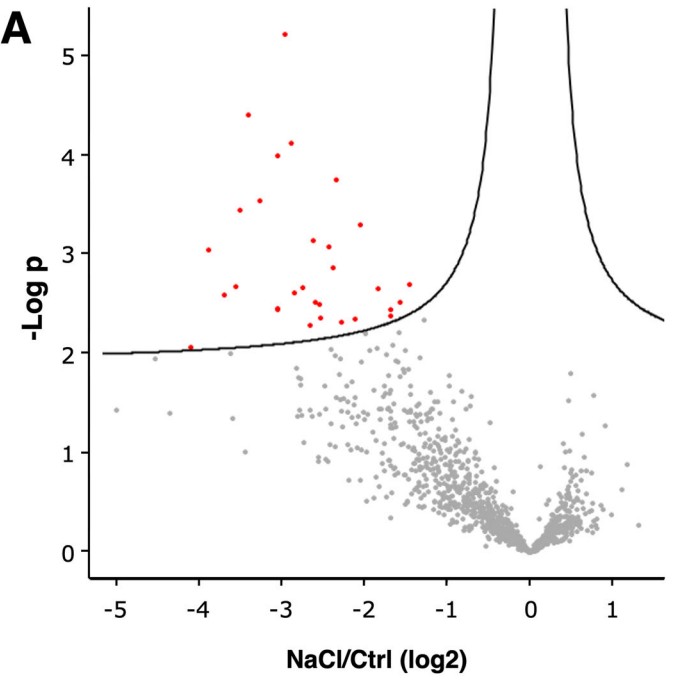

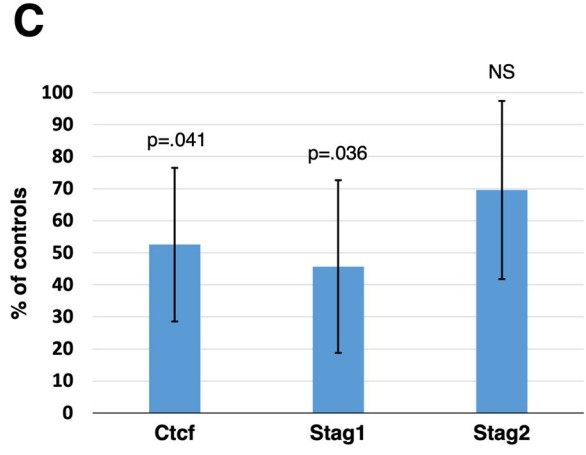

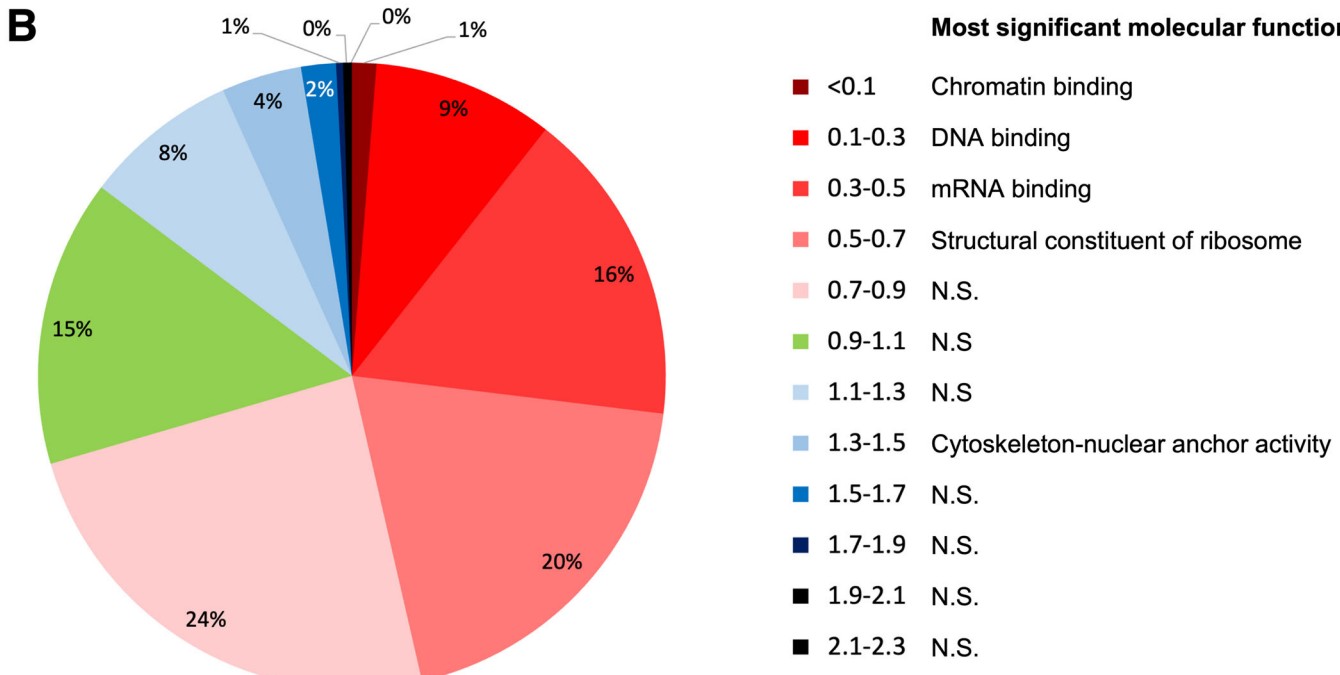

**Most significant molecular function**

- ■ <0.1 Chromatin binding
- ■ 0.1-0.3 DNA binding
- ■ 0.3-0.5 mRNA binding
- ■ 0.5-0.7 Structural constituent of ribosome
- ■ 0.7-0.9 N.S.
- ■ 0.9-1.1 N.S
- ■ 1.1-1.3 N.S
- ■ 1.3-1.5 Cytoskeleton-nuclear anchor activity
- ■ 1.5-1.7 N.S.
- ■ 1.7-1.9 N.S.
- ■ 1.9-2.1 N.S.
- ■ 2.1-2.3 N.S.

the solution was incubated for 5 min on ice. To permeabilize the nuclear membrane, a final concentration of 0.2% Triton X-100 (Thermo Scientific) was added to the suspension. After 5 min of incubation on ice, isolated nuclei were centrifuged as above and resuspended in Iso Buffer (10 mM HEPES pH 7.5, 25 mM KCl, 2 mM MgCl$_2$, 75 mM sucrose, 0.5 mM PMSF, 1 mM DTT)). Nuclei were counted on a hemocytometer.

## Western blotting

To analyze nuclear samples by western blot, nuclei were prepared as follows. MB and MT pellets ($5 \times 10^6$ cells) stored at −80 °C were thawed and their cytoplasmic compartment was depleted by adding 1 ml of Depletion Buffer (10 mM HEPES pH 7.5, 10 mM KCl, 1.5 mM MgCl$_2$, 0.34 M sucrose, 10% glycerol, 0.5 mM EGTA, 0.1%

**Figure 4. Quantitative mass spectrometry of nuclear proteins in nuclei treated or not with NaCl.**

Nuclei isolated from MTs were treated with 0.5 M NaCl or left untreated. Four independent experiments were analyzed separately, then averaged. (A) The diagram shows a dot plot of the variation of 1114 nuclear proteins upon salt treatment, expressed as $\log_2$ (NaCl-treated/untreated). Each dot represents a different protein. The Y axis shows Student's paired $t$ test $P$ values as $\log_2$. (B) The diagram shows the percentages of nuclear proteins whose NaCl-treated/untreated ratios fall within the indicated intervals. Shades of red: ratios <1 (proteins reduced upon NaCl treatment); green: ratios ≈1 (substantially unchanged); shades of blue: >1 (relatively increased). The percentage of all 1114 quantitated proteins in each sector of the graph is shown. For each sector, the most significantly enriched molecular function category (Gene Ontology) is shown next to the legend (Benjamini-Hochberg procedure, FDR < 0.05). N.S. no significant category. (C) Levels of the indicated proteins in salt-treated nuclei, expressed as percentages relative to control nuclei. $P$ values, calculated using Student's paired $t$ test (one-sided), are shown above the columns; NS not significant; $n = 4$ biological replicates. Source data are available online for this figure.

Triton X-100, 1% HALT Protease Inhibitors (Thermo Scientific), 1 mM DTT, 0.5 mM PMSF, 1 mM ATP) and incubating on ice for 30 min. The suspension was then centrifuged at $270 \times g$ for 5 min at 4 °C, the supernatant was discarded and pelleted cells were resuspended in 0.5 ml Depletion Buffer. Identical volumes (60 µl) were drawn from MB and MT suspensions and incubated in Depletion Buffer containing increasing final concentrations of NaCl (0 M, 0.1 M, 0.3 M, 0.5 M, 0.7 M) for 30 min on ice. The suspensions were then centrifuged and the supernatant was collected, while pellets were resuspended in Depletion Buffer.

All samples were sonicated in ice using a Bandelin Sonopuls mini20 homogenizer (20% amplitude, four 5 s pulses with 5 s intervals, for a total of 344 J), then mixed with NuPage LDS Sample Buffer (Invitrogen) and NuPage Sample Reducing Agent (Invitrogen), and heated at 70 °C for 10 min. Proteins were separated on a precast 4–12% polyacrylamide NuPAGE Bis-Tris gel (Invitrogen) in NuPage MES SDS buffer (Invitrogen) at 150 V, then transferred overnight onto a nitrocellulose membrane (Amersham) in transfer buffer (Tris base 20 mM, glycine 150 mM, 20% methanol, pH 8.5) at 40 mA. Membranes were blocked in a TBS-T solution (Tris base 20 mM, NaCl 140 mM, 0.5% Tween 20, pH 7.6) supplemented with 5% nonfat milk (ITW Reagents) for 1 h at RT under shaking. They were then incubated with antibodies to pRb (BD Pharmingen), histone H3 (Abcam), or histone H1 (Santa Cruz Technologies) diluted in TBS-T for 2 h, quickly washed in TBS-T, and incubated with goat anti-mouse IgG (Jackson ImmunoResearch) or goat anti-rabbit IgG (Bio-Rad), conjugated with horseradish peroxidase for 1 h. The membranes were then soaked in Lightwave PLUS ECL solution (GVS) and images captured with an ImageQuant LAS 4000 (Cytiva Life Sciences) apparatus.

## DNA extraction and separation

Total DNA was purified from MBs and MTs with the standard phenol/chloroform method. BrdU-labeled DNA was purified by CsCl density-gradient centrifugation. Briefly, a solution of DNA and CsCl (1.72 g/cm³ density) was ultracentrifuged for 72 h ($238,400 \times g$, +25 °C), using a vertical VTi65 rotor (Beckman Coulter). BrdU-hemisubstituted DNA fractions were identified by refraction index values corresponding to the density range 1.74–1.76 g/cm³ (Luk and Bick, 1977). The DNA was then desalted by dialysis against water, using the Slide-A-Lyzer devices (Thermo Scientific),

## Comparative genomic hybridization

BrdU-DNA was amplified with the WGA2 kit (Merck) and purified using the Genomic DNA Clean & Concentrator (Zymo Research).

Subsequently, DNA from reactivated MTs/proliferating MBs and serum-stimulated quiescent MBs/proliferating MBs was differentially labeled with Cy3/Cy5 (Agilent Technologies), respectively, and hybridized on microarrays for CGH analyses (Mouse Genome CGH Microarray 4x44K; design ID: 015028; Agilent Technologies) following the manufacturer's procedures. Briefly, labeled DNAs were denatured at 95 °C for 3 min and then incubated 30 min at 37 °C. Hybridization was performed at 65 °C for 24 h with rotation in a dedicated G2545A oven (Agilent Technologies). After washing with Agilent wash buffers 1 and 2, dried slides were immediately acquired with a G5761A scanner (Agilent Technologies), imaged with Feature Extraction software (v 8.0, Agilent Technologies) and analyzed using the CytoGenomics software (Edition 2.5.8.11, Agilent Technologies).

## Permeabilization assay

Nuclei were isolated from MBs and MTs as described above. The nuclei were pelleted by centrifuging at $550 \times g$ for 10 min, at 4 °C, and washed once in Isotonic Buffer, incubated for 10 min on ice, centrifuged again, and resuspended in Diffusion Buffer (20 mM HEPES pH 7.5, 110 mM KOAc, 5 mM NaCl, 2 mM MgCl₂, 0.25 M sucrose) to a final concentration of 5000 nuclei/µl. Four volumes of nuclear suspension were then added to one volume of Dextran Mix (0.5 mg/ml fluorescein-isothiocyanate dextran (Sigma-Aldrich), average MW 150 KDa, 0.2 mg/ml DAPI (Sigma-Aldrich), dissolved in Diffusion Buffer), and incubated on ice for 7 min. Samples were spread on glass slides and quickly analyzed by confocal microscopy without fixation. Observations were performed with a Zeiss LSM980 microscope, using a 63×/1.40 NA oil objective and excitation spectral laser lines at 405 and 488 nm. Image acquisition and processing were carried out using the Zeiss confocal software Zen 3.3 (Blue edition) and Adobe Photoshop CS5 software programs (Adobe Systems). Signals from different fluorescent dyes were taken in sequential scan settings. Analysis of treated nuclei was performed using the public domain ImageJ software (v.1.45), quantifying nuclear fluorescence intensity over background intensity to assess the amount of internalized dextran.

## Preparation of *Xenopus laevis* interphase egg extracts (XEE)

Female frogs were injected with 300 U and 600 U of human chorionic gonadotropin 24 h and 16 h before egg collection, respectively, and kept in a 0.1 M NaCl bath. After collection, eggs were washed several times in dejellying buffer (20 mM Tris-HCl pH 8.5, 110 mM NaCl, 5 mM DTT), then rinsed three times with MMR

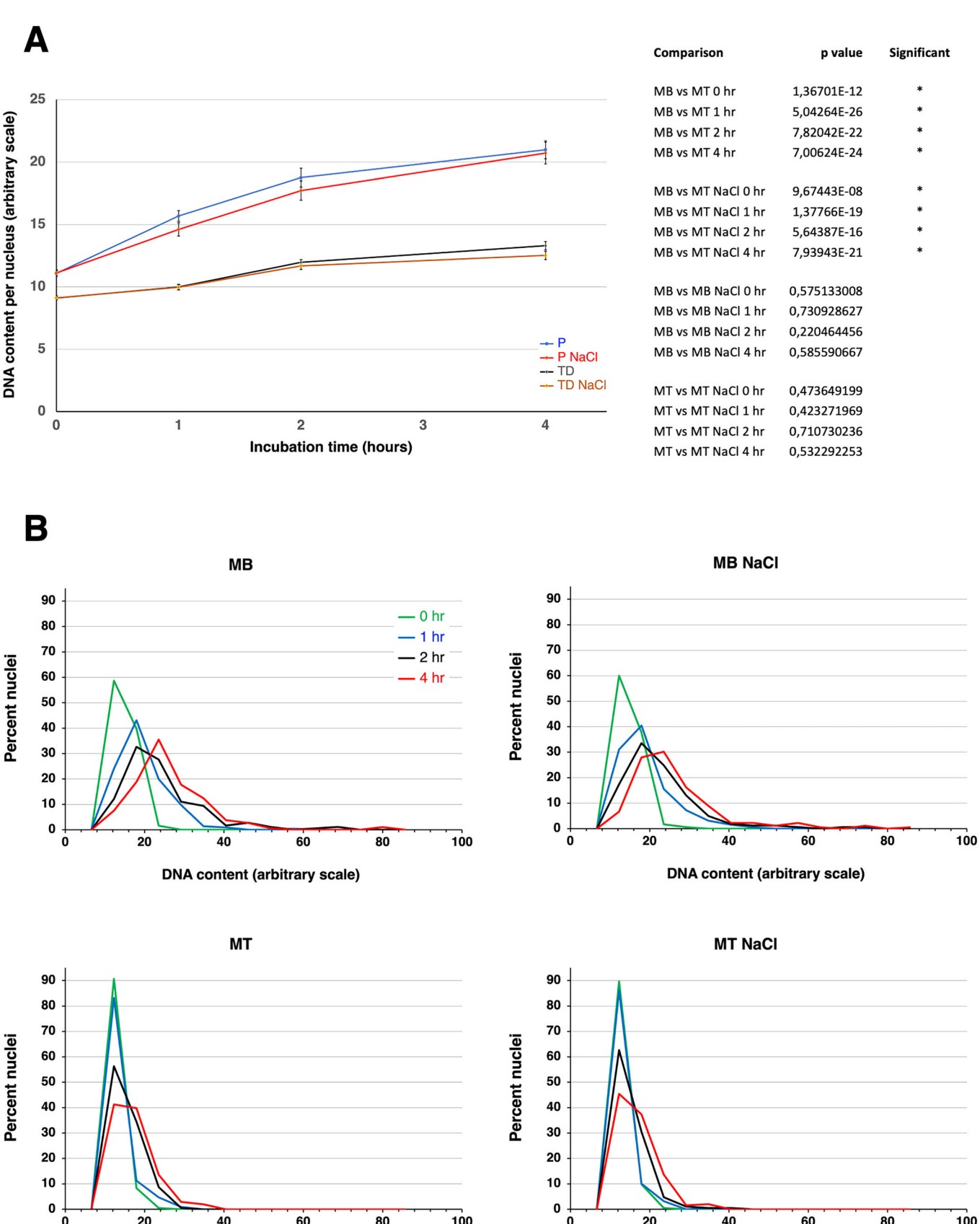

**Figure 5.  Time course of DNA synthesis in MB and MT nuclei treated or not with NaCl.**

Nuclei isolated from MBs or MTs were treated with 0.5 M NaCl or control-treated, then incubated in XEE. At the times shown, aliquots of nuclei were harvested from the incubation mixtures, and stained with DAPI. The total DNA content of about 100 individual nuclei per time point was measured by microfluorimetry. Two independent experiments were carried out and averaged. (A) Median DNA content of the four indicated populations, with standard deviations. MBs nuclei have a higher starting DNA content at 0 hr, because some of them were in S or $G_2$ phase when isolated. MT nuclei, not reactivated, all had a 2n DNA content at 0 h. The table on the right shows comparisons evaluated using a two-sided Student's $t$ test. (B) The DNA content of each of the same four populations is shown in a time-course fashion. Source data are available online for this figure.

Buffer (5 mM HEPES pH 7.5, 100 mM NaCl, 0.5 mM KCl, 0.25 mM $MgSO_4$, 0.5 mM $CaCl_2$, 25 µM EDTA). The dejellied eggs were then incubated with 2 µl of 10 mM calcium ionophore A23187 (Sigma). After 5–10 min, the eggs were washed three times with MMR Buffer and then rinsed twice with ice-cold S Buffer (50 mM HEPES-NaOH pH 7.5, 50 mM KCl, 2.5 mM $MgCl_2$, 250 mM sucrose), supplemented with 2 mM β-mercaptoethanol and 15 µg/ml leupeptin just before use. The activated eggs thus obtained were packed by centrifugation (a few seconds at 3300 × g in a microcentrifuge) to get rid of excess buffer and then crushed by centrifugation at 16,100 × g for 10 min. The crude extract was separated from yolk and insoluble material, supplemented with 40 µg/ml of cytochalasin B and then ultracentrifuged for 18 min at 70,000 rpm at 4 °C with a TLA100 rotor (Beckman). The final extract, obtained by mixing the clarified protein extract and the membrane fraction, was supplemented with 200 µg/ml cycloheximide (Calbiochem).

## Salt treatment of nuclei

Pellets of MB or MT nuclei (typically, $1-2 \times 10^5$ in 5 µl), stored at −80 °C, were thawed, resuspended, and mixed 1:1 with a 1 M NaCl solution, to a final concentration of 0.5 M NaCl. They were then incubated on ice for 10 min. After the incubation, nuclei were diluted in 1 ml of Iso Buffer and centrifuged 3 min at 1100 × g. The nuclei were then immediately used in XEE experiments.

For mass spectrometry analyses, freshly isolated nuclei were washed once in Iso Buffer, suspended in 50 mM ammonium bicarbonate, and sonicated as described under "Western blotting".

## XEE-mediated DNA replication assay

In different in vitro replication assays, we measured newly synthesized DNA in the nuclear population as a whole or total DNA content at the level of individual nuclei. In some experiments, the percentage of nuclei that underwent DNA replication was estimated.

### Newly synthesized DNA
Frozen XEE aliquots (30 µl) were thawed on ice and promptly added with 2 µl of 20x energy mix (20x energy mix: 0.2 mg/ml creatine phosphokinase (Sigma-Aldrich), 2 mg/ml phosphocreatine (Sigma-Aldrich), 20 mM $MgCl_2$, 2 mM EGTA), 2 µl of $^3$H-dTTP, and 6–8 µl of nuclei in suspension (about 2000–10,000 nuclei, depending on the preparation), for a total volume of 40 µl. The reaction was incubated at 23 °C. At different time points, 4 µl aliquots of the mix were harvested, mixed with 16 µl of Stop Buffer (80 mM Tris-HCl pH 8, 8 mM EDTA, 0.1% $H_3PO_4$, 5% SDS, 50% glycerol, 0.025% bromophenol blue, 1 mg/ml proteinase K, 200 µg/ml RNAse), and incubated overnight at 37 °C.

The digested samples were transferred onto glass microfilters (Whatman GF/C 2.5) and allowed to dry in open air. The filters were then washed in a cold solution of 5% trichloroacetic acid (TCA) and 2% inorganic pyrophosphate (PPi) for at least 20 min at 4 °C, washed four times in 5% cold TCA, and finally once with 100% ethanol. The filters were transferred into vials with a scintillation cocktail, and radioactivity was measured with a Beckman LS 6000 SC liquid scintillation counter.

### Total DNA content
Nuclei (~90,000) were incubated as described above, except that $^3$H-dTTP was omitted. Harvested aliquots were fixed and stained by mixing them with an equal volume of a solution containing 50% glycerol, 3.7% formaldehyde, and 0.2 µg/ml DAPI in Buffer A (45 mM PIPES pH 7.2, 45 mM NaCl, 240 mM KCl), placed onto a microscope slide, and overlaid with a coverslip. Images of random microscopic fields containing several nuclei were captured using an Axioskop 2 Plus fluorescence microscope equipped with a Neofluar ×40 objective (NA 0.75), and an AxioCam camera (Zeiss). The images were analyzed with the ImageJ software (v. 1.43 u). DAPI fluorescence intensity, linearly proportional to DNA content, was measured in all well-defined individual nuclei in the captured images, with local background subtraction. The experimenter performed blinded measurements of image intensities.

### Percentage of DNA-replicating nuclei
To calculate the percentage of nuclei that synthesized DNA in XEE, parallel incubations were set up exactly as above, except that the label was 0.3 µM Cy3-dCTP. Harvested nuclei were scored as Cy3-positive or -negative using a fluorescence microscope.

### Purified DNA replication
Purified DNA replication experiments were carried out as described above, except that the label was biotin-16-dUTP (Roche, 1 mM; 2 µl per reaction). DNA purified from either MBs or MTs (500 ng) was added to each reaction. Triplicate aliquots of the reactions, harvested at different incubation times, were transferred onto a membrane (Optitran BA-S 85, 0.45 µm) using a dot blot microfiltration apparatus (Bio-Rad). The membrane was washed three times with 0.16 M HCl. It was then crosslinked at 80 °C for 2 h and incubated with 1% bovine serum albumin (BSA; Sigma) for 1 h. Streptavidin-peroxidase from *Streptomyces avidinii* (Sigma; 1:10,000), conjugated to horseradish peroxidase (HRP), was used to detect biotinylated DNA. The membrane was subsequently incubated with WesternBright ECL substrate and imaged using the ImageQuant LAS 4000 digital imaging system (GE Healthcare). Densitometric analyses were performed using the ImageJ software (NIH).

## Transmission electron microscopy

Purified nuclei were fixed in 2.5% glutaraldehyde, 2% paraformaldehyde, and 2 mM calcium chloride in 0.1 M Na-cacodylate buffer (pH 7.4) overnight at 4 °C. Subsequently, samples were washed in buffer, post-fixed in 1% $OsO_4$ in 0.1 M Na-cacodylate for 1 h at room temperature, and treated with 1% tannic acid in 0.1 M Na-cacodylate buffer for 30 min. Post-fixed pellets were rinsed in the same buffer, dehydrated in 20% ethanol solution for 10 min and in block stained with UA-Zero EM Stain (Agar Scientific Ltd, UK) for 1.5 h. The samples were then dehydrated through a graded series of ethanol solutions (up to 100%) and then embedded in medium-hardness epoxy resin (Agar100 Resin Kit, Agar Scientific Ltd). The samples were incubated for polymerization at 65 °C for 60–72 h. Ultrathin sections, obtained with a LEICA EM UC6 ultramicrotome (Leica Microsystems), were examined at 100 kV with a Philips EM208S TEM (Thermo Fisher Scientific) equipped with a Megaview II SIS camera (Olympus).

## Mass spectrometry

Proteins extracted from nuclei treated or not with 0.5 M NaCl were analyzed by LC-MS/MS after cysteine reduction with 1 mM tris(2-carboxyethylphosphine) (TCEP, Sigma-Aldrich) and alkylation with 12 mM iodoacetamide (Sigma-Aldrich), and overnight digestion with trypsin (1.6 ng/µl, enzyme /protein ratio 1/50 w/w, Promega) at 37 °C. Peptides were then injected into an Ultimate 3000 UHPLC (Dionex, Thermo Fisher Scientific) in line with an Orbitrap Fusion Tribrid mass spectrometer (Thermo Fisher Scientific). After a desalting step in an Acclaim PepMap 100 C18 trap column (Thermo Fisher Scientific), analytical separations were performed on a single 45 cm-long silica capillary (Silica Tips FS 360-75-8, New Objective) packed in-house with a C18, 1.9 µm, 100 Å resin (Michrom Bioresources) and heated at 45 °C in an oven (Sonation). A 90 min-long HPLC gradient was used, combining buffer B (95% acetonitrile and 0.1% formic acid) with Buffer A (5% acetonitrile and 0.1% formic acid) as follows: from 5% to 6% B in 5', to 32% in 65', and up to 80% in 10'. The Data Independent Acquisition (DIA) mode was used. MS data were acquired in the 300–1200 $m/z$ range split into 40 contiguous windows with 1 $m/z$ overlap, fragmented by HCD 28%, and the MS/MS spectra were acquired in the Orbitrap at 30 K resolution in the 145–1450 $m/z$ range.

Protein identification was obtained using the SwissProt mouse database containing the manually curated list of histones (24,945 sequences) (El Kennani et al, 2017). The raw data were analyzed with the DIA-NN software (v. 1.8.1) (Demichev et al, 2020). Statistical analyses were performed using the Perseus software (v. 2.0.11) (Tyanova et al, 2016). Data analyses regarded only nuclear proteins, based on SwissProt annotations, while contaminating non-nuclear proteins were ignored. To evaluate significant changes in protein abundances between NaCl-treated and -untreated samples, raw intensity values were normalized to total core histone values. After normalization, all values were transformed to $\log_2$. Only proteins quantified in at least five of eight samples were considered. Missing values were imputed, replacing them with values from the normal distribution (Width 0.3, Down Shift 1.8). Significance was evaluated using Student's $t$ test (FDR 0.05, S0 = 0.1, 250 randomizations). Functional Gene Ontology categories were assigned using the David Bioinformatics Functional annotation tool.

## Data availability

The datasets produced in this study are available in the following database: All the mass spectrometry data have been deposited to the ProteomeXchange Consortium via the MassIVE partner repository with the MSV000096839 dataset identifier ftp://MSV000096839@massive-ftp.ucsd.edu.

The source data of this paper are collected in the following database record: biostudies:S-SCDT-10_1038-S44319-025-00554-x.

## Peer review information

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

## Acknowledgements

We thank Alessia Errico and Vincenzo Sannino for generously providing *Xenopus* egg extracts. We thank our colleague Francesca Spadaro of the Core Facilities of the Istituto Superiore di Sanità for her help with confocal microscopy. We are also grateful to Silvia Soddu for critically reading the manuscript. This work has been supported in part by grant ECS00000024 "Rome Technopole" (2022-2025) funded by European Union – NextGeneration EU through the Italian Ministry of University and Research under PNRR – "Ecosistemi di Innovazione", Mission 4, Component 2, Investment 1.5. MC and DP were supported by grants no. PE-2011-02346905 and RF-2010-2310869, respectively, from Ministero della Salute, Italy. SC received a grant "Bando Ricerca Indipendente ISS 2021" from Istituto Superiore di Sanità, Italy (project code ID ISS20-101042c64ab6). VC received funding from Associazione Italiana per la Ricerca sul Cancro, Italy under IG 2023—ID. 28725 project.

## Author contributions

**Sara Zribi**: Data curation; Formal analysis; Validation; Investigation; Methodology. **Gladio Joel Giannitelli**: Data curation; Validation; Investigation; Writing—original draft. **Laura Bernardini**: Data curation; Methodology. **Federica Censi**: Data curation; Methodology. **Serena Camerini**: Conceptualization; Data curation; Formal analysis; Investigation; Methodology. **Lucia Bertuccini**: Formal analysis; Investigation; Methodology. **Francesca Iosi**: Data curation; Investigation; Methodology. **Emilia Giannella**: Data curation; Formal analysis; Investigation. **Massimo Sanchez**: Data curation; Formal analysis; Methodology. **Elisa Consoli**: Investigation; Methodology. **Andrea Casagrande**: Methodology. **Alessandra Cenci**: Data curation; Formal analysis; Methodology. **Giovanna Floridia**: Methodology. **Antonio Novelli**: Methodology. **Alessandro Giuliani**: Software; Formal analysis; Methodology. **Vincenzo Costanzo**: Conceptualization; Resources; Formal analysis. **Marco Crescenzi**: Conceptualization; Data curation; Formal analysis; Supervision; Funding acquisition; Investigation; Methodology; Writing—original draft; Project administration; Writing—review and editing. **Deborah Pajalunga**: Conceptualization; Data curation; Formal analysis; Investigation; Methodology.

Source data underlying figure panels in this paper may have individual authorship assigned. Where available, figure panel/source data authorship is listed in the following database record: biostudies:S-SCDT-10_1038-S44319-025-00554-x.

## Disclosure and competing interests statement

The authors declare no competing interests.

# Expanded View Figures

**Figure EV1.   Reactivation of DNA replication in MBs and MTs.**

(**A**) MTs reactivated for the CGH experiments were continuously labeled with BrdU until harvest. Percentages of BrdU$^+$ MT are shown. (**B**) MBs made quiescent in suspension culture were either not replated (negative control) or replated in tissue culture dishes and pulse-labeled with BrdU for 30 min at the indicated times. Percentages of BrdU$^+$ MBs are shown. (**C**) Examples of immunofluorescence micrographs on which the data presented are based. Bars: 25 μm.

▶

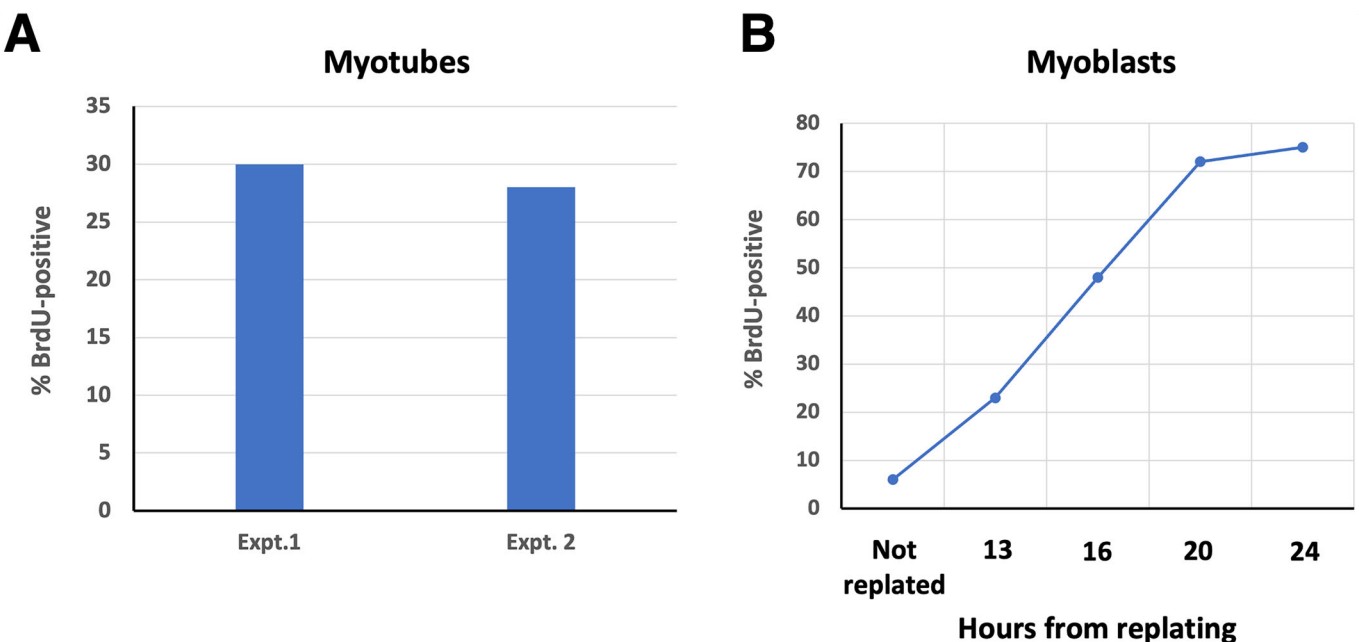

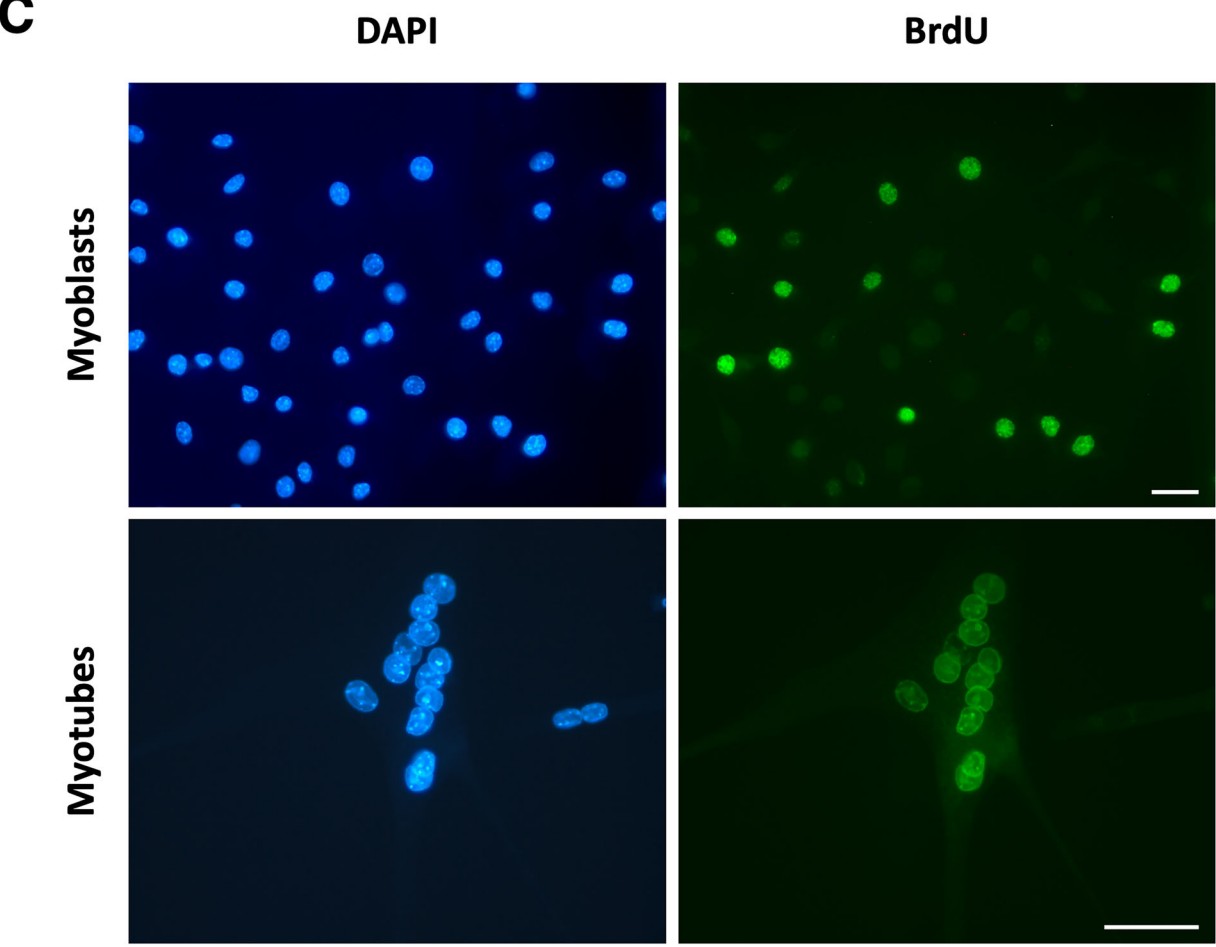

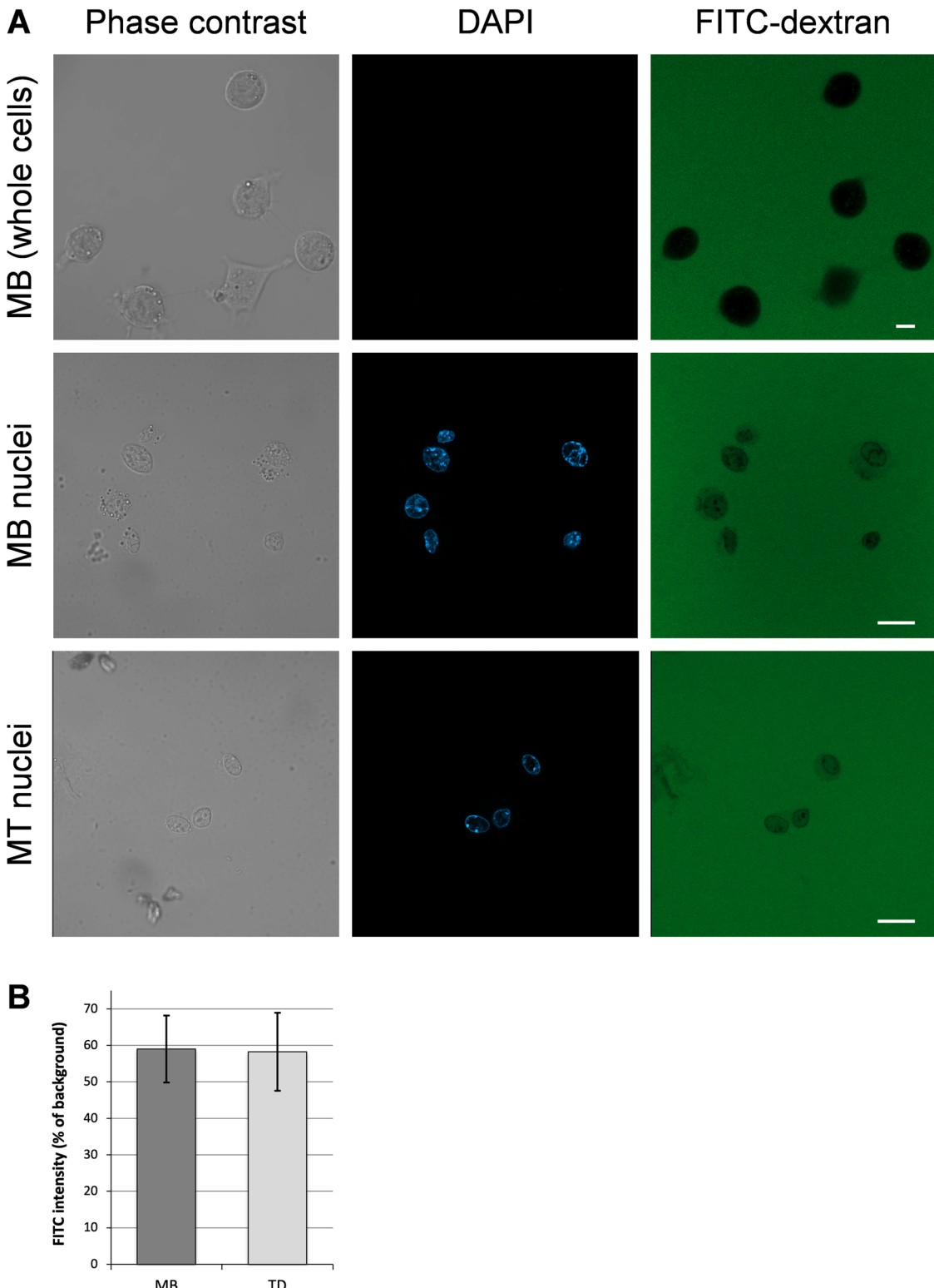

**Figure EV2. Permeabilization assays.**

(A) Whole, non-permeabilized MBs (negative control), MB nuclei, or MT nuclei were incubated in a DAPI/FITC-dextran mix, spread on glass slides and analyzed by confocal microscopy. A set of representative images is shown for each sample. (B) Dextran uptake in MB and MT nuclei, expressed as percentage of background fluorescence intensity, with standard deviation. Statistical analysis indicates no significant difference between the two samples MBs=59.0 ($n = 21$); MTs=58.2 ($n = 19$); unpaired $t$ test, $P = 0.814$).

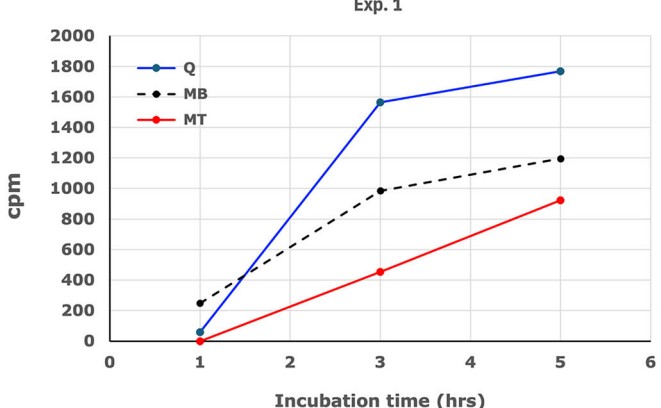

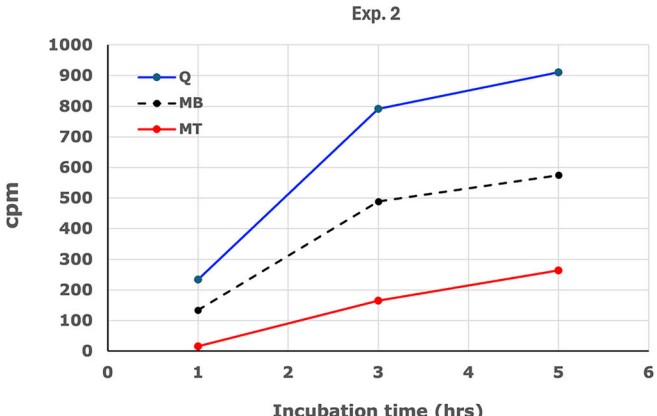

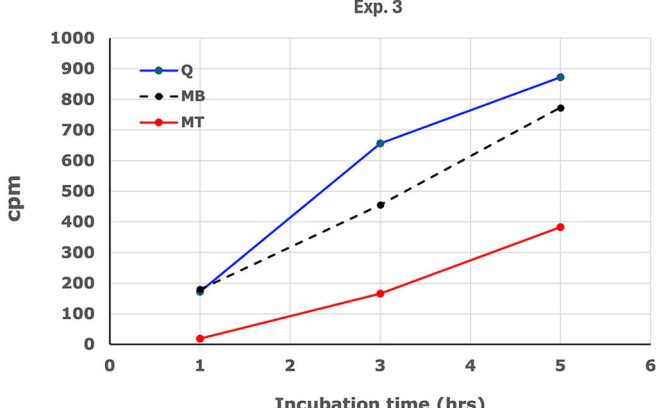

**Figure EV3.  ³H TTP in MB and MT nuclei incubated in XEE.**

Incorporation of ³H-dTTP into nuclei incubated in XEE. The three experiments averaged in Fig. 2 are shown here separately. Disintegration counts per minute (cpm) were measured at the indicated incubation times and normalized to the number of nuclei in each reaction.

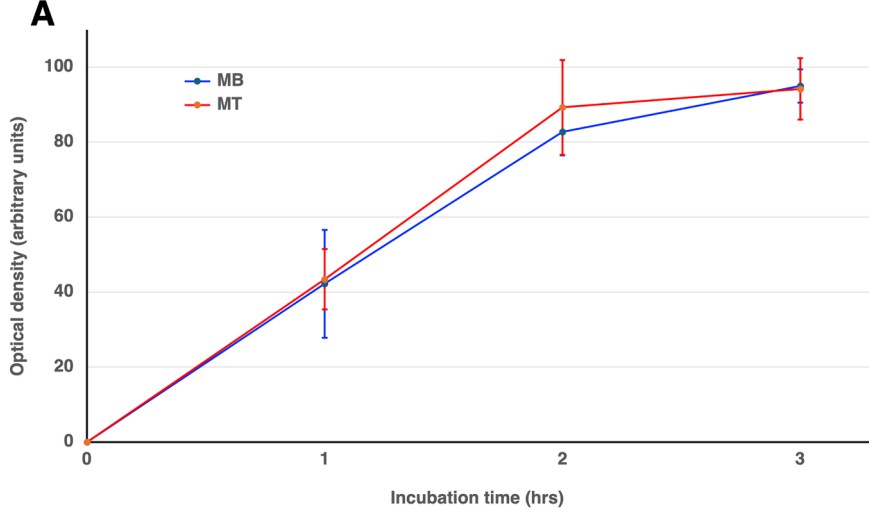

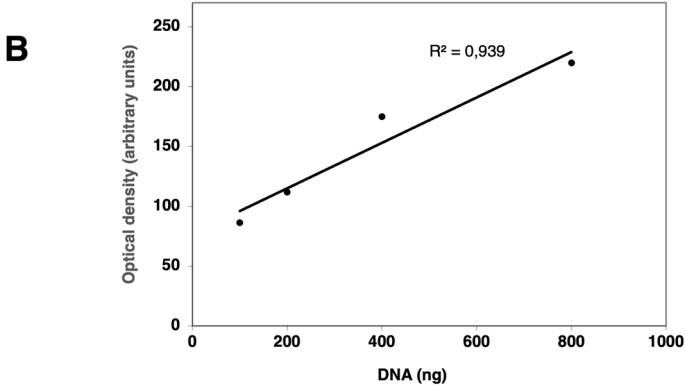

**Figure EV4. Replication of purified DNA in XEE.**

(A) Incorporation of biotin-16-dUTP into DNA purified from MBs or MTs in XEE. The average results of three independent experiments are shown, with standard deviations. Label incorporation was detected by dot blot, using horseradish peroxidase-conjugated streptavidin (see Methods and Protocols for details). The amount of DNA synthesized at the indicated times is expressed as the average optical density of three replicate spots from the same aliquot on the dot blot. (B) To ensure the accuracy and quality of the dot blots, each included a linear regression based on varying amounts of PCR-synthesized biotinylated DNA. An example from the second experiment is shown here. In the three experiments, the coefficient of determination ($R^2$) ranged from 0.9390 to 0.9737.

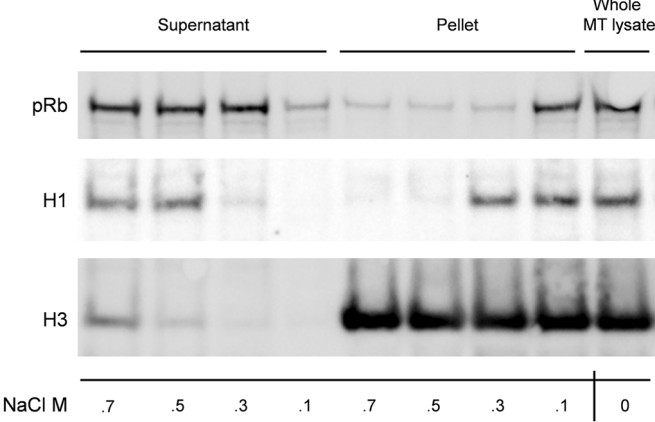

**Figure EV5. NaCl concentration finder.**

Nuclei isolated from MT were treated for 30 min with the indicated molar concentrations of NaCl and centrifuged at 500 g. The indicated proteins from supernatants and nuclear pellets were analyzed by western blotting. pRb: Retinoblastoma protein; H1: histone H1 (all subtypes); H3: histone H3 (all subtypes).

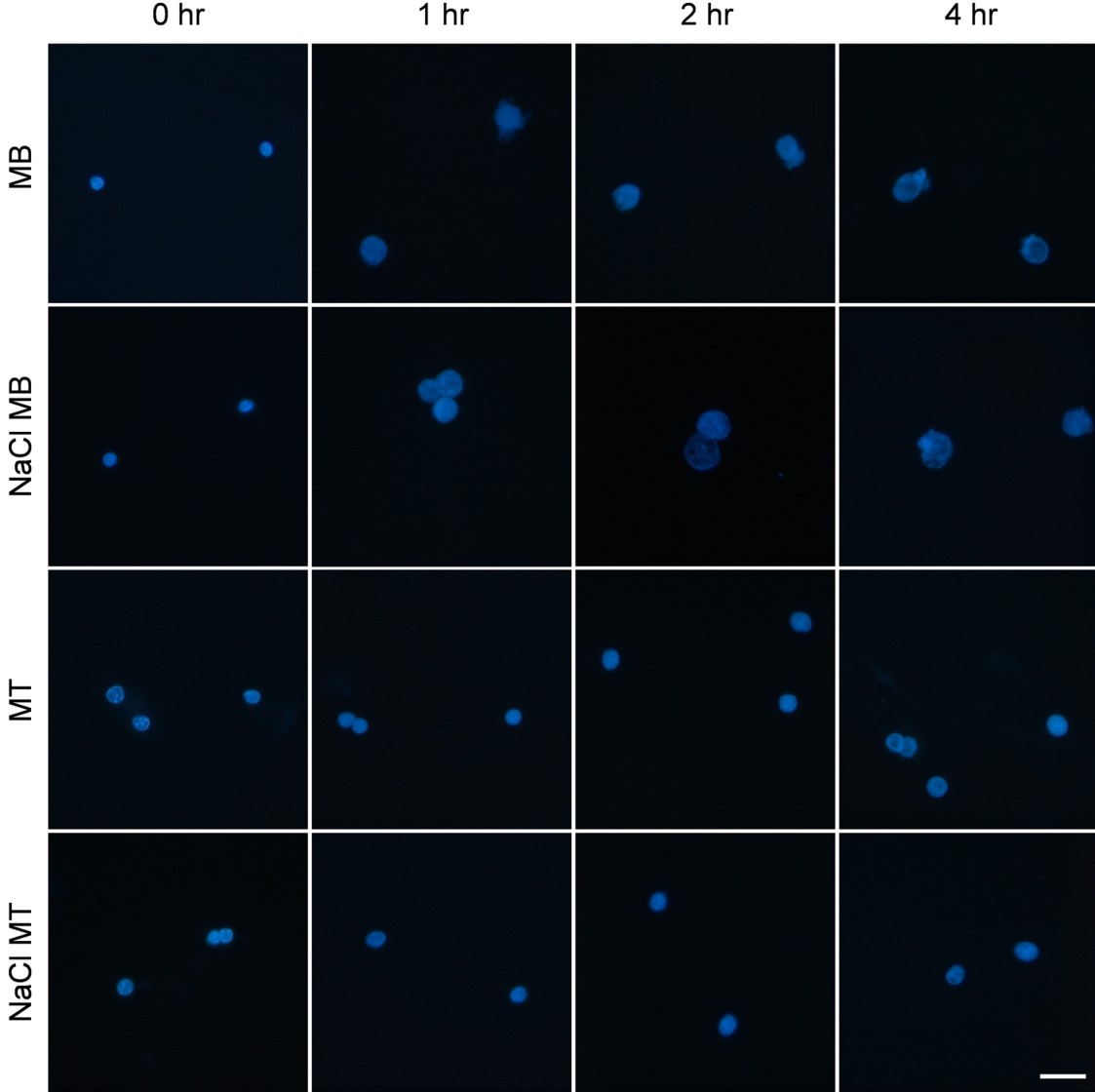

**Figure EV6. Microscopy of MB and MT nuclei incubated in XEE.**

Fluorescent microscopy photographs of MB and MT nuclei, treated or not with NaCl, incubated in XEE for the indicated times, and stained with DAPI. These are examples of the nuclei used for the microfluorimetric measurements shown in Fig. 5 and Fig. EV3. Bar: 25 μm.

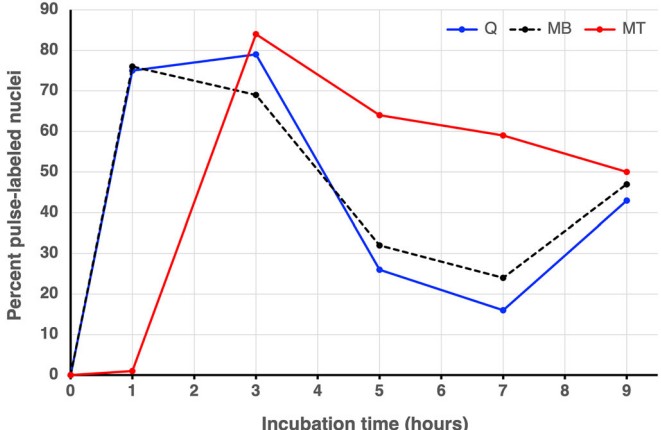

**Figure EV7.  Time course of pulse-labeled nuclei in XEE.**

Nuclei from quiescent (Q) or proliferating MBs (MB) and non-reactivated myotubes (MT) were incubated in XEE in parallel. Thirty minutes before times 1, 3, 5, 7, and 9 h, one incubation mixture per nucleus type was pulse-labeled with Cy3-dCTP for 30 min before times 1, 3, 5, 7, and 9. The graph shows the percentages of Cy3-positive nuclei at each time point. Continuous Cy3 labeling throughout the experiment of an identical reaction set (three reactions: Q, MB, and MT) showed that all nucleus types were reactivated similarly: at the 5 h time point, Cy3-positive nuclei were: Q, 96%; MB, 95%; MT, 93%.

