## [Peer Review File · EMBO Reports]

Structural obstruction to full DNA replication in terminally differentiated skeletal muscle cells

Sara Zribi, Gladio Joel Giannitelli, Laura Bernardini, Federica Censi, Serena Camerini, Lucia Bertuccini, Francesca Iosi, Emilia Giannella, Massimo Sanchez, Elisa Consoli, Andrea Casagrande, Alessandra Cenci, Giovanna Florida, Antonio Novelli, Alessandro Giuliani, Vincenzo Costanzo, Marco Crescenzi, and Deborah Pajalunga

Corresponding authors: Marco Crescenzi (marcosilvia86@alice.it)

Review Timeline:

Submission Date:	30th Jan 25
Editorial Decision:	11th Mar 25
Revision Received:	17th Jun 25
Editorial Decision:	22nd Jul 25
Revision Received:	28th Jul 25
Accepted:	5th Aug 25

Editor: Esther Schnapp

Transaction Report:

Dear Marco,

Thank you for the submission of your manuscript to EMBO reports and for your proposed revision plan. I discussed all with my colleagues here and we agree that the revised ms would make a nice contribution to our exploratory reports section, given that the data are not fully conclusive.

I would thus like to invite you to revise your manuscript to the best of your abilities with the understanding that the referee concerns must be fully addressed and their suggestions taken on board. Please address all referee concerns in a complete point-by-point response. Acceptance of the manuscript will depend on a positive outcome of a second round of review. It is EMBO reports policy to allow a single round of major revision only and acceptance or rejection of the manuscript will therefore depend on the completeness of your responses included in the next, final version of the manuscript.

We realize that it is difficult to revise to a specific deadline. In the interest of protecting the conceptual advance provided by the work, we recommend a revision within 3 months (11th Jun 2025). Please discuss the revision progress ahead of this time with the editor if you require more time to complete the revisions.

- 1) A data availability section providing access to data deposited in public databases is missing. If you have not deposited any data, please add a sentence to the data availability section that explains that.
- 2) Your manuscript contains statistics and error bars based on $n=2$. Please use scatter blots in these cases. No statistics should be calculated if $n=2$.

5) a complete author checklist, which you can download from our author guidelines . Please insert information in the checklist that is also reflected in the manuscript. The completed author checklist will also be part of the RPF.

6) Please note that all corresponding authors are required to supply an ORCID ID for their name upon submission of a revised manuscript (. Please find instructions on how to link your ORCID ID to your account in our manuscript tracking system in our Author guidelines

7) Before submitting your revision, primary datasets produced in this study need to be deposited in an appropriate public database (see <https://www.embopress.org/page/journal/14693178/authorguide#datadeposition>). Please remember to provide a reviewer password if the datasets are not yet public. The accession numbers and database should be listed in a formal "Data Availability" section placed after Materials & Method (see also <https://www.embopress.org/page/journal/14693178/authorguide#datadeposition>). Please note that the Data Availability Section is restricted to new primary data that are part of this study. * Note - All links should resolve to a page where the data can be

accessed. *

- the name of the statistical test used to generate error bars and P values,
- the number (n) of independent experiments (please specify technical or biological replicates) underlying each data point,
- the nature of the bars and error bars (s.d., s.e.m.),
- If the data are obtained from n {less than or equal to} 2, use scatter blots showing the individual data points.

12) All Materials and Methods need to be described in the main text using our 'Structured Methods' format, which is required for all research articles. According to this format, the Methods section includes a Reagents and Tools Table (listing key reagents, experimental models, software and relevant equipment and including their sources and relevant identifiers) followed by a Methods and Protocols section describing the methods using a step-by-step protocol format. The aim is to facilitate adoption of the methodologies across labs. More information on how to adhere to this format as well as a downloadable template (.docx) for the Reagents and Tools Table can be found in our author guidelines:

An example of a Method paper with Structured Methods can be found here: <https://www.embopress.org/doi/full/10.1038/s44320-024-00037-6#sec-4>

I look forward to seeing a revised form of your manuscript when it is ready.

Best regards,

Esther

Referee #1:

In this manuscript the authors investigate the cause of the incompetence of terminally differentiated, postmitotic myotubes to fully reactivate the cell cycle. By comparing myotube and myoblast nuclei incubated with replicative *Xenopus* egg extract, which provides the DNA replication machinery, the authors describe essential differences in timing and length of genome replication. While myoblast nuclei attained complete DNA replication, myotube nuclei could not duplicate more than half of their genomes. Genomic hybridization of newly synthesized DNA showed that the replicative failure occurs stochastically. Finally, by disassembling and disorganizing chromatin with strong salt treatment they conclude that such incompetence is rooted in the core structure of chromatin in myotube nuclei.

This work provides a significant extension of a long-lasting effort from this group toward understanding the biology of the irreversible withdrawal from the cell cycle typical of postmitotic myotubes. Indeed, this group has made the major contribution on this specific topic and the results shown here add interesting mechanistic insights that definitely deserve to be published.

The data convincingly support the author's conclusions, and I have no major issues on the experimental procedures and analysis.

My only recommendation to the authors is to address more accurately the issue related to the conclusion that the inability of myotubes to reactivate the cell cycle should be ascribed primarily to structural obstacles, based on the salt treatment and the assumption that it removes most chromatin-bound proteins. Indeed, it is possible that this treatment does not remove chromatin-bound proteins implicated in the formation and maintenance of higher order chromatin interactions that define nuclear domains and the 3D genome organization.

The authors might want to experimentally address this point or simply critically discuss it.

Referee #2:

The work by Zribi et al addresses reasons why the DNA in the nuclei of postmitotic muscle cells is prevented from replication. The authors suggest that it is the chromatin structure that obstructs replication.

While the question is important, the paper is very preliminary both in terms of the experimental design and the presentation of the data. Most figure legends lack sufficient information to understand the data. My most important critic is that there is almost no data specifically relevant to chromatin, which is the main claim of the manuscript.

Below I provide some comments that I hope the authors will find useful.

- Show in a quantitative manner that rMTs actually fall short of DNA replication in the experimental setting used in this paper. Figure 2 is indicative of this but not sufficient.
- Give details about purity of MTs, nucleus isolation, assurance that the lack of difference is not due to selective enrichment of MB nuclei in the transfected cells. What was the transfection efficiency of myonuclei specifically?
- "We reasoned that we should reduce the protein content of both MT and MB nuclei, as far as they retained their differences in DNA replicability" This sentence is very unclear. How does protein content relate to differences in DNA replicability? There is also a complete lack of specificity here.
- The authors claim in the first part of the manuscript that it is the interaction between proteins that is disrupted by the salt treatment. But later they say it is the actual protein content that is affected. One would expect the same content even if the interactions are disrupted. If the total content is relevant, how is the 83% relate to other non-histone proteins? The authors write that 83% of the linker H1 was displaced from chromatin but how exactly was this assayed? How does the mass-spec on total nuclear extract able to be a measure of displacement?
- The sentence "indicating that most nuclear proteins whose content was reduced by salt play no role in restricting DNA replication in MTs" The entire experimental design is based on salt treatment that they show reduces salt content. Thus, writing this sentence undermines in my view even the limited relevance of the experiments.

Referee #3:

In this manuscript, Zribi and colleagues set-out to address the key question of what makes postmitotic muscle cells reluctant to undergo DNA synthesis. This is a relevant question for basic biology with implications in regeneration studies. They perform a series of neat experiments to show that the limitation to full replication is not functional but somehow structural. I have some suggestions to improve the manuscript:

1. There is an apparent discordance between the limited genomic under-replication detected by CGH on unique sequences (Figure 1) and the severe reduction in DNA synthesis rate and replication completion detected by *Xenopus* extracts assays (Figure 2). Since in the later approximation address the whole genome, including repetitive sequences that account for a vast part of it, it will be interesting investigating the replicative status of repetitive elements. This could be inferred by sequencing total DNA along time from the extracts and using the latest genome annotation based on long reads (see also point 3).
2. The authors argue that salt-resistant specific nucleosomes might impede DNA replication at MTs. Can they detect differences in the enrichment of modified nucleosomes in their salt-treated MS data between MT and MB cells? Additional insights of the differentially enriched proteins should be presented.
3. Another possible explanation for the MT cells resistance to full replication could be that postmitotic cells have re-arranged chromosomes/chromatin regions that block the replication of such regions. Is there any evidence on this? Further, the authors could check which regions are being underreplicated in salt-treated nuclei by total DNA sequencing of time 4h vs 0h, for example. Comparing those results with similar analyses in non-treated nuclei (point 1) will make more robust their conclusion that the hindrance to fully replicate relies on structural properties of chromatin.

Minor points:

1. More information can be added to some of the figures to make them more appealing and easy to interpret to the readers. For example, in Fig2 a panel with a scheme of the experimental set-up can be included, or in Fig 1C, adding below the positions of the E/L replicated regions mapped on this cell line (or a similar type).
2. Fig 1: Please show a flow cytometry control or similar showing that MTs treated with siRNA against p21 and p27 indeed enter the cell cycle. Same for quiescent MBs reactivation.
3. Fig2, Y axis: % of replication
4. Annotation of the meaning of the color-code in Fig 4B should be added, alongside with specific examples of proteins in each category

Structural obstruction to full DNA replication in terminally differentiated skeletal muscle cells

Point-by-point response to the reviewers' comments

We are submitting a revised version of our manuscript EMBOR-2025-61247, now entitled "Structural obstruction to full DNA replication in terminally differentiated skeletal muscle cells". The title has been modified in response to the referees' remarks. Please find here below our point-by-point responses.

General considerations

From several of the referees' remarks, we recognize that we did not convey our message clearly enough. The manuscript has therefore been thoroughly revised, with particular attention to the points raised by the reviewers.

All three referees objected to our references to chromatin. Although it is difficult to imagine that the obstacle we describe resides elsewhere, we acknowledge that we have little direct evidence for this. We have therefore softened all allusions to chromatin, including in the title.

In the following point-by-point response, we have included the referees' remarks (in italic) and assigned sequential numbers to them for easy reference.

In the new version of the manuscript, all revisions appear in red.

Referee 1:

1. My only recommendation to the authors is to address more accurately the issue related to the conclusion that the inability of myotubes to reactivate the cell cycle should be ascribed primarily to structural obstacles, based on the salt treatment and the assumption that it removes most chromatin-bound proteins. Indeed, it is possible that this treatment does not remove chromatin-bound proteins implicated in the formation and maintenance of higher order chromatin interactions that define nuclear domains and the 3D genome organization. The authors might want to experimentally address this point or simply critically discuss it.

1R. We thank this referee for their positive comments.

Our conclusion about the presence of a structural obstacle derives directly from the evidence that MTs cannot fully replicate their DNA even in *Xenopus* extracts. As we explained in the last paragraph of the Introduction and in the relevant paragraph of the Results, since the extract contains the whole DNA replication machinery, it is indifferent to any functional defects of myotubes, but still sensitive to physical hurdles. Salt treatment is not essential for reaching this conclusion; rather, it helps to narrow down the search for the molecular underpinnings of the phenomenon.

Nonetheless, the referee is correct: we cannot state as a fact that higher order chromatin domains are disrupted by salt. Demonstrating this point would require performing Hi-C, a

technique currently not available in our laboratory, which we could not establish during the revision. However, in the revised manuscript, we have highlighted (page 6) that two key proteins for loop formation, CTCF and Stag1, are reduced by 50% in salt-treated nuclei (new Panel C in Fig. 4). Yet, such reduction has no influence on DNA replication in myotube nuclei. The implications are discussed on page 7, where we also point out that the typical size of DNA loops is larger than the resolution of our CGH. If 3D genome organization were involved in limiting DNA replication, we should have detected hyporeplicated DNA regions in myotubes. We would like to emphasize the evidence from electron microscopy, which reveals a severe disruption of nuclear organization morphology, again with no consequence on defective DNA replication.

Referee 2:

2. *While the question is important, the paper is very preliminary both in terms of the experimental design and the presentation of the data.*

2R. We thank this reviewer for recognizing the importance of the question we address and for their constructive comments.

While we acknowledge that we do not yet have a molecular specification of the physical obstacle we describe, to our knowledge this work provides the first evidence of such a structural impediment. We believe this concept has the potential to shift current perspectives on the postmitotic state. We specifically chose this Journal because it “focuses on novelty and the physiological and/or functional significance of a finding, rather than the level of mechanistic detail reported” (Author Guidelines).

We believe that most of the concerns raised stem from our insufficient clarification of key aspects of the manuscript. We have strived to improve the manuscript, as detailed below.

3. *Most figure legends lack sufficient information to understand the data.*

3R. We recognize that several figure legends lacked sufficient detail and have revised most of them to improve clarity (legends to Figs. 1, 2, 4, and 5 and EV Figs. 1–3).

3. *My most important critic is that there is almost no data specifically relevant to chromatin, which is the main claim of the manuscript.*

3R. This is correct. As noted in the general considerations at the beginning of this response, we have removed the reference to chromatin from the title and reduced emphasis on chromatin all over the paper.

4. Show in a quantitative manner that rMTs actually fall short of DNA replication in the experimental setting used in this paper. Figure 2 is indicative of this but not sufficient.

4R. We assume the criticism concerns rMT *nuclei*, rather MT as cells, because the reviewer refers to Fig. 2. The data presented in Fig. 2 are indeed quantitative. As indicated in the legend, they are normalized to 100%, representing the highest radiolabel incorporation obtained in each experiment. This normalization is necessary due to the inherent variability of these experiments, arising from factors such as XEE batch potency, radioactive label freshness, and the precise number of nuclei used. We have now added Appendix Fig. 2,

showing separate graphs for the three experiments summarized in Fig. 2, with actual disintegration cpm.

5. *Give details about purity of MTs, nucleus isolation, assurance that the lack of difference is not due to selective enrichment of MB nuclei in the transfected cells. What was the transfection efficiency of myonuclei specifically?*

5R. In the Methods and Protocols section, under "Cells and cell cultures", the differentiation index is stated to be >95%.

We have now added that transfection efficiency is 100% if determined with fluorescent siRNAs. However, MT reactivation shows interexperimental variation (range: 25-80%). In the legend to Fig. 1 we now state that MT reactivation efficiency was 30% in the first experiment and 28% in the second. We underscore that percentages of reactivation do not affect the results, because we isolated BrdU-hemisubstituted DNA, thus considering only reactivated MTs. Full nuclear isolation details are provided under "Isolation of nuclei".

As to the possibility of selective enrichment of MB nuclei, we suspect there has been a misunderstanding. With "lack of difference" the reviewer seems to refer to the CGH data shown in Fig. 1, specifically Panels B and C. However, CGH experiments were performed with DNA extracted from cultured cells, not isolated nuclei. We have made this clearer in the description of the experiments (page 4). Selective enrichment in MBs is highly unlikely, given the starting purity of MT cultures (>95%). Further, there is no significant MT death due to reactivation, at the time rMTs are harvested.

Despite the above, in case we have misinterpreted the question and selective loss of MT nuclei remains a concern, we have performed two reconstruction experiments, mixing known numbers of MT and MB nuclei. Setting the recovery of MT nuclei in each experiment to 100%, the recovery of MB nuclei was 71% and 80%, respectively. Thus, if anything, there was some depletion of MB nuclei, rather than an enrichment. This is due to the fact that nucleus isolation conditions have been optimized for MTs, which are more robust cells. We do not think it is necessary to show these data but, if the reviewer requests, we can add them to the Appendix.

6. *"We reasoned that we should reduce the protein content of both MT and MB nuclei, as far as they retained their differences in DNA replicability" This sentence is very unclear. How does protein content relate to differences in DNA replicability? There is also a complete lack of specificity here.*

6R. We apologize for the lack of clarity. We meant to say that our aim was to simplify the system as much as possible, thereby facilitating the analysis of its molecular constituents and ultimately the identification of the molecular determinants of incomplete DNA replication in MTs. However, system simplification can proceed only as far as the difference between MBs and MTs is preserved; if it is lost, the implication is that we have removed its molecular determinants. We have rephrased the rationale to clarify this point (page 5).

7. *The authors claim in the first part of the manuscript that it is the interaction between proteins that is disrupted by the salt treatment. But later they say it is the actual protein content that is affected. One would expect the same content even if the interactions are disrupted. If the total content is relevant, how is the 83% relate to other non-histone proteins? The authors write that 83% of the linker H1 was displaced from chromatin but how exactly*

was this assayed? How does the mass-spec on total nuclear extract able to be a measure of displacement?

7R. We apologize for neglecting to remind the reader, at this point, that the nuclei are permeabilized. A protein detached from chromatin (or the nuclear scaffold) is lost in the washes (cfr. Fig. EV2 on this point, pellets vs. supernatants). All of the reviewer's questions can be answered in light of this. We have revised the manuscript accordingly (page 5).

8. *The sentence "indicating that most nuclear proteins whose content was reduced by salt play no role in restricting DNA replication in MTs" The entire experimental design is based on salt treatment that they show reduces salt content. Thus, writing this sentence undermines in my view even the limited relevance of the experiments.*

8R. We assume the reviewer meant "reduces protein content". The reasoning here proceeds by subtraction: since the salt treatment does not affect the difference in DNA replicability, proteins whose levels are reduced by salt can be presumed *not* to be involved in restricting DNA replication in MTs. We have clarified this point on page 7.

Referee 3:

In this manuscript, Zribi and colleagues set-out to address the key question of what makes postmitotic muscle cells reluctant to undergo DNA synthesis. This is a relevant question for basic biology with implications in regeneration studies. They perform a series of neat experiments to show that the limitation to full replication is not functional but somehow structural. I have some suggestions to improve the manuscript:

9. *There is an apparent discordance between the limited genomic under-replication detected by CGH on unique sequences (Figure 1) and the severe reduction in DNA synthesis rate and replication completion detected by Xenopus extracts assays (Figure 2). Since in the later approximation address the whole genome, including repetitive sequences that account for a vast part of it, it will be interesting investigating the replicative status of repetitive elements. This could be inferred by sequencing total DNA along time from the extracts and using the latest genome annotation based on long reads (see also point 3).*

9R. We thank this reviewer for their appreciative comments and constructive attitude. The discordance is indeed only apparent. In our CGH, we load equal amounts of BrdU-hemisubstituted DNA from both rMTs and reference MBs. The flat lines shown in Fig. 1B,C do not indicate that the two cell types replicate their DNA to the same extent, but that all regions are similarly represented in the two samples. In fact, MTs require roughly twice as many nuclei to achieve similar signal levels, due to their lower BrdU incorporation. Put differently: these results show equal probability of replication across all regions, not equal amounts of replicated DNA per nucleus. We realize that we did not make this clear in the original manuscript. We have now modified the CGH paragraph to make it more explanatory (page 4). We have also modified the schematic presented in Fig. 1A, to make the experiment easier to interpret graphically.

We have evidence, generated from cultured MTs by NGS, showing that all repetitive and non-repetitive sequences are replicated to the same extent. Such data will be published elsewhere. See also the response to point 11 (reviewer's point 3).

10. *The authors argue that salt-resistant specific nucleosomes might impede DNA replication at MTs. Can they detect differences in the enrichment of modified nucleosomes in their salt-treated MS data between MT and MB cells? Additional insights of the differentially enriched proteins should be presented.*

10R. This is our ultimate goal, but it requires very complex mass spectrometry analyses. We must leave this crucial point to future studies. We have made this explicit in the final discussion (page 8).

11. *Another possible explanation for the MT cells resistance to full replication could be that postmitotic cells have re-arranged chromosomes/chromatin regions that block the replication of such regions. Is there any evidence on this? Further, the authors could check which regions are being underreplicated in salt-treated nuclei by total DNA sequencing of time 4h vs 0h, for example. Comparing those results with similar analyses in non-treated nuclei (point 1) will make more robust their conclusion that the hindrance to fully replicate relies on structural properties of chromatin.*

11R. In fact, there is evidence to the contrary. We have stressed that the block to DNA replication is uniformly distributed and that all DNA regions are replicated with the same probability in rMTs. The mechanism proposed above implies that some regions should be replicated definitely less than others or not at all and, in our opinion, is not compatible with the CGH results.

Despite our reservations about the premises of this request, in the revision plan submitted to the editors, we stated that we would try to perform the experiment proposed by the reviewer. We emphasized that it would be technically challenging. However, it would have addressed the question of whether the *Xenopus* extract experiments and the *in vivo* CGH measurements are probing the same phenomenon (we strongly believe so, supported by their qualitative and quantitative similarities: cf. Figs. 2B and 5).

Unfortunately, circumstances beyond our control that we have disclosed to the editors prevented us from attempting the experiment. However, we would like to emphasize that it is not critical for the paper, whose main conclusion — that there exists a physical impediment to full DNA replication — is, in our opinion, convincingly demonstrated.

Minor points:

1. *More information can be added to some of the figures to make them more appealing and easy to interpret to the readers. For example, in Fig2 a panel with a scheme of the experimental set-up can be included, or in Fig 1C, adding below the positions of the E/L replicated regions mapped on this cell line (or a similar type).*

1R. The schematic has been added as Panel A of Fig. 2. The E/L replication map of chromosome 1 in foetal myoblasts has been added as Panel D of Fig. 1, using a different style to emphasize that it represents a reference, not newly generated data.

2. *Fig 1: Please show a flow cytometry control or similar showing that MTs treated with siRNA against p21 and p27 indeed enter the cell cycle. Same for quiescent MBs reactivation.*

2R. Flow cytometry cannot be applied to MTs, as they are very large and multinucleated. However, we have added Appendix Fig. 1, which shows: (a) the percentages of reactivated (BrdU⁺) MTs just before harvest for DNA extraction; (b) the progressive entry of MBs into S phase following replating; and (c) exemplary evidence on which the previous two panels are based.

3. *Fig2, Y axis: % of replication*

3R. Corrected into "% DNA replication".

4. *Annotation of the meaning of the color-code in Fig 4B should be added, alongside with specific examples of proteins in each category*

4R. We have annotated the color code to improve readability. Adding examples might have been somewhat arbitrary, potentially giving disproportionate weight to the specific proteins or categories chosen. Although the full list of nuclear proteins, complete with quantitative data, is included as Supplementary Table 1, we have tried to address the reviewer's request in a uniform and objective manner: we are showing the most representative functional GO category in each segment of the pie, where applicable.

Dear Marco,

Thank you for the submission of your revised manuscript. We have now received the enclosed report from referee 1, who was asked to assess it, and I am happy to say that this referee supports the publication of your study now. Only a few editorial requests will need to be addressed before we can proceed with the official acceptance of your manuscript:

- Please add the corresponding author's email address to the title page.
- Individual callouts for Figure 2 (A-B), 4AB are missing, please add. EV Table 1 is cited but missing (perhaps this is Suppl. Table 1?), please clarify.
- The Dataset EV1 file is now part of the source data (SD) for Figure 4. If this file is exclusively SD, then the label Dataset EV1 should not be used; if the file needs to be provided individually, then it needs to be uploaded as a Dataset and called out as such. Please correct. We can also discuss this further if you have any questions or comments.
- Given that the Appendix file only has 3 figures I would like to suggest that you change these figures into EV figures. It is OK to have 7 EV figures. Please correct all file names and callouts accordingly.
- The labels of the EV figures in the individual files need correction to Figure EV1, etc. (instead of EV Figure 1, etc.)
- Please upload the Reagents and Tools table as a separate file and remove it from the ms file.
- Please carefully go through the SD checklist again, as there is inconsistency between the SD checklist and the SD provided, which needs to be amended.
- Please upload higher resolution figures for EV1A Microscopy, EV2 Blots and EV4 Microscopy. Under evaluation these figures are pixelated and while not a cause for concern, it can sometimes give the impression of image alteration to critical readers.

* Figure Legends - Comments *

- Please note that the measure of center for the error bars needs to be defined in the legend of figure EV1 B
- Please note that scale bar and its definition are missing for figure EV1 A.

I made a few minor changes to the abstract that needs to be written in present tense. Please let me know whether you agree with this:

Terminal cell differentiation is often associated with permanent withdrawal from proliferation, termed the postmitotic state. Though widespread among vertebrates and determinant for their biology, the molecular underpinnings of this state are poorly understood. Postmitotic skeletal muscle myotubes can be induced to reenter the cell cycle; however, they generally die as a result of their inability to complete DNA replication. Here, we explore the causes of such incompetence. Genomic hybridization of newly synthesized DNA shows that the replicative failure does not concern specific genomic regions, but can stochastically affect any of them. Myoblast and myotube nuclei are incubated in replicative *Xenopus* egg extract, which provides a full DNA replication machinery. While myoblast nuclei attain complete DNA replication, those from myotubes, even in these conditions, duplicate less than half of their genomes, strongly indicating that the structure of myotube chromatin obstructs DNA replication. Furthermore, disassembling and disorganizing chromatin with a strong salt treatment does not modify the replicative differences between the two nuclei types, suggesting that they are rooted in the core structure of chromatin.

- The synopsis image at its final size of 550 pixels wide has too small text. Please send us a new image at the correct size with readable text, thank you.

I slightly modified and shortened your short summary and bullet points. Do you agree with this:

Postmitotic myotubes cannot complete DNA replication following forced cell cycle reactivation. Comparative genomic hybridization and *Xenopus* egg extracts show that this is primarily due to a diffuse physical obstacle to replication.

- In myotubes forced to re-enter the cell cycle, all DNA regions are similarly and randomly underreplicated, suggesting the presence of a uniformly diffuse obstacle that stochastically impedes DNA replication.
- Myotube nuclei show the same incomplete DNA replication when incubated in *Xenopus* egg extracts, which provide a full

replication machinery, indicating that the obstacle is structural.

- Reduction of nuclear protein levels by salt treatment, combined with mass spectrometry analyses, hints at a role for core chromatin proteins.

Referee #1:

The authors have replied to the reviewers' concerns with a very reasonably revised version of the manuscript, which now conveys more clearly the findings related to the postmitotic state of skeletal myotubes and its conceptual novelty.

The authors addressed the remaining editorial issues.

Prof. Marco Crescenzi
Italian National Institute of Health
Core Facilities
Viale Regina Elena, 299
Via G.B. Manzella, 146
Rome 00161
Italy

Dear Marco,

I am very pleased to accept your manuscript for publication in the next available issue of EMBO reports. Thank you for your contribution to our journal.
